



# ih-RIDME: a pulse EPR experiment to probe the heterogeneous nuclear environment

Sergei Kuzin[1], Victoriya N. Syryamina[2], Mian Qi[3], Moritz Fischer[3], Miriam Hülsmann[3], Adelheid Godt[3], Gunnar Jeschke[1], and Maxim Yulikov[1]

[1]Department of Chemistry and Applied Biosciences, ETH Zurich, Vladimir-Prelog-Weg 2, 8093 Zurich, Switzerland
[2]Voevodsky Institute of Chemical Kinetics and Combustion, Institutskaya str. 3, Novosibirsk 630090, Russia
[3]Faculty of Chemistry and Center for Molecular Materials (CM$_2$), Bielefeld University, Universitätsstraße 25, 33615 Bielefeld, Germany

**Correspondence:** Sergei Kuzin (skuzin@ethz.ch)

**Abstract.** Intermolecular hyperfine relaxation-induced dipolar modulation enhancement experiment (ih-RIDME) is a pulse EPR experiment that can be used to probe the properties of a nuclear spin bath in the vicinity of an unpaired electron. The underlying mechanism is the hyperfine spectral diffusion of the electron spin during the mixing block. A quantitative description of the diffusion kinetics being applied to establish the ih-RIDME data model allows to extend this method for systems with heterogeneous nuclear arrangements assuming a distribution of the local nuclear densities. The heterogeneity can stem from the solvent or the intrinsic nuclei of structurally flexible (macro)molecule. Therefore, the fitted distribution function can further serve for heterogeneity characterization, quantification and structure-based analysis. Here, we present a detailed introduction to the principles of the ih-RIDME application to heterogeneous systems. We discuss the spectral resolution, determination of the spectral diffusion parameters and influence of noise in the experimental data. We further demonstrate the application of the ih-RIDME method to a model spin-labeled macromolecule with unstructured domains. The fitted distribution of local proton densities was reproduced with the help of the Monte-Carlo-generated conformational ensemble. Finally, we discuss several pulse sequences exploiting the HYperfine Spectral Diffusion Echo MOdulatioN (HYSDEMON) effect with an improved signal-to-noise ratio.

## 1 Introduction

Among pulse EPR experiments there are two big families of techniques which can be summarized under the names 'hyperfine spectroscopy' and 'electron-electron dipolar spectroscopy'. Pulse electron-electron dipolar spectroscopy (PDS) techniques have a longer upper distance limit and can probe electron spin pairs separated by even beyond 10 nm. Thus, PDS techniques are considered as 'long-range structure' determination EPR techniques. Contrary, it is common to view hyperfine spectroscopy techniques in pulse EPR as 'local structure' determination techniques, mainly targeting magnetic nuclei with resolved hyperfine splitting, within the shell of not more than 1 nm.(Dikanov and Tsvetkov, 1992) The hyperfine techniques can further address nuclei beyond the 1 nm distance limit provided their homonuclear interaction is negligible.(Asanbaeva et al., 2023; Seal et al., 2023; Judd et al., 2022; Gauger et al., 2024)



In the case of many magnetic nuclei present in the vicinity of a paramagnetic centre, those weakly coupled cannot be well resolved in the NMR spectrum forming a so-called matrix peak positioned at the corresponding nuclear Zeeman frequency.

The intensity of the matrix peak can be used as a measure of the total number of weakly coupled nuclei, e.g. in the water accessibility experiments.(Volkov et al., 2009; Noethig-Laslo et al., 2004)

In many samples, especially related to biological systems or organic materials, there is a large number of magnetic nuclei of the same sort, typically protons, coupled to each other by dipolar and exchange interaction. Such a network of coupled protons is called a 'proton spin bath' and it undergoes continuous quasi-stochastic local fluctuations of the spin states of

individual protons. This process, called proton spin diffusion, covers regions rather close to the paramagnetic centres present in the sample. As a result of proton spin diffusion, electron spin senses a fluctuating hyperfine field formed by nearby protons. Accordingly, the resonance frequency of the electron spin fluctuates within the related hyperfine spectrum. This process is called electron spectral diffusion. At short electron-proton distances the proton spin diffusion is blocked by the gradient of the electron's magnetic field. Thus, the hyperfine spectrum relevant for the electron spectral diffusion only includes weak

couplings to those protons beyond the blocking limit for the nuclear spin diffusion. All the protons within the blocked range only contribute to the full hyperfine spectrum of a given electron spin as a shift but do not participate in the electron spectral diffusion process.

The concept of a blocking threshold is a simplification – there is a smooth transition between completely blocked protons and protons with unperturbed spin diffusion. The largest contribution to the electron spectral diffusion comes from the protons

close to the blocking conditions. The currently available evaluations suggest that the electron spectral diffusion process is sensitive to the protons in a range that may start between 0.5 and 1 nm and may end between 1.5 nm and 3 nm, depending on typical strengths of proton-proton couplings which, in turn, are concentration-dependent.

Due to the existence of a blocked range, in the proton sub-ensemble relevant to the electron spectral diffusion, there are usually no proton couplings that are way stronger than others. Thus, the corresponding sub-ensemble of protons can be ap-

proximated as a quasi-continuous 'magnetic cloud'. Accordingly, it is justified to ask about the shape of such a cloud and its magnetic density.

The proton spin diffusion/electron spectral diffusion process manifests in the electron spin echo dephasing, often being the main contribution to the phase memory time of paramagnetic centres. Similarly, the proton spin diffusion/electron spectral diffusion process can be detected in the stimulated electron spin echo experiment and its variations. The stimulated electron

spin echo experiments exist in the form of a three-pulse electron spin echo envelope modulation (ESEEM) technique and modifications thereof, as well as the relaxation-induced dipolar modulation enhancement (RIDME, Figure 1a) technique and its variants. In the former class, the length of the longitudinal block is incremented, and the transverse part is usually kept constant. In the RIDME-based class, the increment relation is the opposite which allows to avoid the contribution of the electron spin longitudinal relaxation to the primary data. Thus, we focus on the determination of the parameters of the electron

spectral diffusion/nuclear spin diffusion via the RIDME pulse sequence and we refer to this experiment in this context as an intermolecular hyperfine (ih-) RIDME.(Kuzin et al., 2022, 2024a)





The ih-RIDME traces, previously discussed in the literature as a RIDME background,(Astashkin, 2015) feature shape sensitivity to the local proton density and the mixing time (Figure 1b). This allows the RIDME experiment to be exploited to probe the proton environment of the paramagnetic site. The dependence on the mixing time is related to the average strength

of nuclear-nuclear interaction, hence, reflects the properties of the homonuclear coupling network. As a step further, the developed quantitative model for ih-RIDME enables the characterisation of heterogeneous systems in terms of the distribution function of local proton densities. The ih-RIDME technique thus offers intermediate-range structural information, beyond the arrangement of ligands and solvent molecules in the coordination shell and/or in the first two-three solvation shells (which is typical information from other hyperfine spectroscopy techniques). This feature makes ih-RIDME the first hyperfine-based

structural method for long-range structure determination (albeit the sensitive distance range for ih-RIDME is still substantially shorter than for PDS techniques). While global data fitting in ih-RIDME significantly improves the fit stability, it also raises a question about optimizing the signal quality and measurement time.

This article is organized as follows: after introducing the theory for ih-RIDME signals from anisotropic proton distributions, we give an account of the spectral resolution. Next, we discuss the influence of the noise on the fit of ih-RIDME data, followed

by a detailed discussion of the procedure for determining fit parameters of the ih-RIDME kernel in the heterogeneous distribution case. After this, we turn to analyze the experimental ih-RIDME data obtained for a model compound containing one nitroxide unit with a highly anisotropic distribution of protons. Finally, before turning to conclusions, we give an account of selecting different versions of the RIDME experiment for better performance of ih-RIDME measurements.

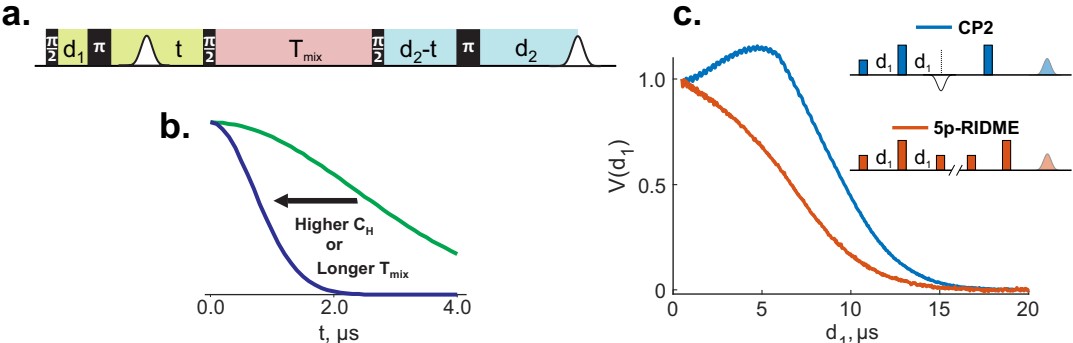

**Figure 1.** (a) ih-RIDME pulse sequence. The pulse labels indicate their flip angles. Highlighted blocks correspond to the preparation (lime), diffusion (light red) and the detection (light blue) functional elements of the sequence. (b) ih-RIDME traces become steeper if local nuclear concentration ($C_H$) or the length of the diffusion block ($T_{mix}$) increase. (c) Comparison of the echo intensity in Carr-Purcell sequence (CP2) and 5p-RIDME sequence as the function of the first interpulse delay ($d_1$) measured in a protonated solution. The echo in CP2 has a maximum due to the dynamical decoupling effect in the presence of proton-proton coupling. In the case of RIDME, the echo monotonically decreases which features a dissipative effect of the longitudinal block.



## 2 Theoretical background

The mixing block in the ih-RIDME experiment is an essential functional element. This can be demonstrated in a comparison of echo evolution in the Carr-Purcell-2 sequence (CP2), which is also the observer sequence in the four-pulse and five-pulse double electron-electron resonance (DEER) experiment, and the five-pulse RIDME experiment when the length of the first interpulse delay is incremented (Figure 1c). Note that such an increment is unconventional for the RIDME experiment. In the CP2 experiment, the echo intensity is of non-monotonic behaviour with a distinct maximum (blue line). This effect is attributed to the

presence of protons.(Bahrenberg et al., 2021; Jeschke, 2023) During the CP2 experiment, electron magnetization continuously evolves in the transverse plane. The RIDME sequence can be seen as a CP2 sequence with an inserted $\pi/2 - T_\mathrm{mix} - \pi/2$ block, called mixing block, during which the electron magnetization is aligned in the $z$-direction. In the example in Figure 1c the mixing block is inserted at the position of the primary echo. As a result of such modification, the echo intensity monotonically decreases as a function of $d_1$ (orange line). This demonstrates that the mixing block has a strong influence on the echo intensity.

Besides, the observed intensity decay is indicative of dissipative properties of spin dynamics during the mixing block and this justifies the term "spectral diffusion" introduced in this Section.

### 2.1 Longitudinal spectral diffusion

The ih-RIDME signal is a decay of the echo intensity in the RIDME experiment when the position of the mixing block shifts in the pulse sequence. The decay occurs due to the longitudinal spectral diffusion. This effect arises when the spectroscopic

properties of the electron spin before and after the mixing block are not strictly correlated leading to incomplete refocusing of spin-spin interactions. In ih-RIDME, the interaction of interest is the hyperfine interaction with close nuclei. The Hamiltonian of this interaction in the secular approximation is

$$\hat{H}_\mathrm{hf} = \sum_{j=1}^{N} A_j \hat{S}_z \hat{I}_{z,j} \tag{1}$$

where $A_j$ are the hyperfine coupling constants and $N$ is the number of nuclei. The main example of nuclei in this work is the

protons because they are the most abundant nuclei in applications such as biological studies and dynamic nuclear polarization (DNP) and have the largest gyromagnetic ratio, thus, their spin-spin interaction is the strongest. For protons, the driving force for the nuclear state evolution is the homonuclear dipolar coupling

$$\hat{H}_\mathrm{dd} = \sum_{j \neq k} \omega_{j,k} \left( \frac{1}{2} \hat{I}_{z,j} \hat{I}_{z,k} - \frac{1}{4} \hat{I}_j^+ \hat{I}_k^- \right) \tag{2}$$

where $\omega_{j,k}$ are the nuclear dipolar coupling constants. The nuclear Zeeman Hamiltonian commutes with both $\hat{H}_\mathrm{hf}$ and $\hat{H}_\mathrm{dd}$ and

is, therefore, not considered in the following discussion.

The ih-RIDME pulse sequence (Figure 1a) has three functional elements: preparation (highlighted in lime), mixing (light red) and detection blocks (light blue). We assume for generality that the spin Hamiltonians during these blocks are $\hat{H}^\mathrm{(prep)}$, $\hat{H}^\mathrm{(diff)}$ and $\hat{H}^\mathrm{(det)}$ respectively. By denoting the corresponding free evolution propagators as $U_\mathrm{prep}$, $U_\mathrm{diff}$ and $U_\mathrm{det}$, the full expression





for the ih-RIDME signal after propagation of the electron spin coherence $\hat{S}_x$ reads

$$V(t) \propto \mathrm{tr}(\hat{S}_x \hat{\sigma}_3) \tag{3}$$

where

$$\hat{\sigma}_3 = U_{\mathrm{det}}(d_2)U_\pi U_{\mathrm{det}}(d_2 - t)\hat{\sigma}_2 U_{\mathrm{det}}(d_2 - t)^\dagger U_\pi^\dagger U_{\mathrm{det}}(d_2)^\dagger \tag{4}$$

$$\hat{\sigma}_2 = U_{\pi/2}U_{\mathrm{diff}}(T_{\mathrm{mix}})U_{\pi/2}\hat{\sigma}_1 U_{\pi/2}^\dagger U_{\mathrm{diff}}(T_{\mathrm{mix}})^\dagger U_{\pi/2}^\dagger \tag{5}$$

$$\hat{\sigma}_1 = U_{\mathrm{prep}}(d_1 + t)U_\pi U_{\mathrm{prep}}(d_1)\hat{S}_x U_{\mathrm{prep}}(d_1)^\dagger U_\pi^\dagger U_{\mathrm{prep}}(d_1 + t)^\dagger. \tag{6}$$

Analytical evaluation of Eq. (3) is not feasible for a large number of nuclei and full Hamiltonian, i.e. when $\hat{H}^{(\mathrm{prep})} = \hat{H}^{(\mathrm{diff})} = \hat{H}^{(\mathrm{det})} = \hat{H}_{\mathrm{hf}} + \hat{H}_{\mathrm{dd}}$. In a simplified theoretical picture, we consider the hyperfine interaction and neglect the nuclear-nuclear interaction during the preparation and detection part of the RIDME pulse sequence

$$\hat{H}^{(\mathrm{prep})} = \hat{H}^{(\mathrm{det})} = \hat{H}_{\mathrm{hf}}. \tag{7}$$

Such an approximation means that the nuclear magnetic state during preparation and detection is static and it allows us to characterize the interaction of the electron spin with the nuclear reservoir by the so-called hyperfine field (in angular frequency units)

$$\omega_{\mathrm{hf}} = \sum_{j=1}^{N} A_j m_{I,j} \tag{8}$$

which we assume to be conserved during the electron spin transverse evolution. The hyperfine field is the resonance offset of the electron spin due to interaction with all the nuclei. This value depends on both nuclear coordinates via $A_j$ and on the magnetic state of the nuclei via $m_{I,j}$ and is thereby distributed.

We consider some properties of the hyperfine field. The distribution of the hyperfine fields is called a hyperfine spectrum $\rho(\omega_{\mathrm{hf}})$. This function describes the distribution of resonance offsets in an ensemble of electron spins with fixed nuclear coordinates but different nuclear spin states. The hyperfine spectrum also depends on the nuclear arrangement around the electron spin. Its mean value is not detectable in the pulse experiment because such a constant offset is refocused at the echo detection. In the statistical limit, we approximate the shape of the hyperfine spectrum as Gaussian

$$\rho(\omega_{\mathrm{hf}}) \approx \frac{1}{\sqrt{2\pi}\sigma} \exp\left(-\frac{(\omega_{\mathrm{hf}} - \overline{\omega})^2}{2\sigma^2}\right). \tag{9}$$

Given that the offset $\overline{\omega}$ is not detectable, the standard deviation $\sigma$ is sufficient to represent the hyperfine spectrum.

We can relate $\sigma$ to some geometric properties of a homogeneous nuclear reservoir. Assuming that all nuclei with $R < r < +\infty$ are included in the ensemble we find

$$\sigma^2(\infty) - \sigma^2(R) = \frac{4\pi}{3}B^2\frac{C_{\mathrm{H}}}{R^3} \tag{10}$$



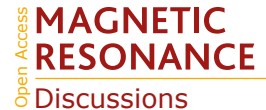

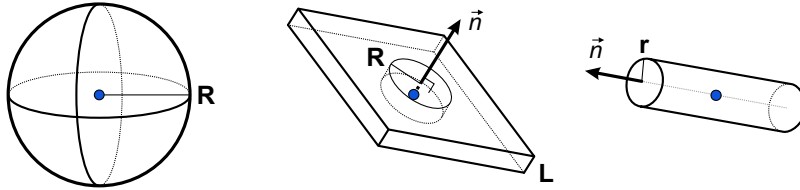

**Figure 2.** From left to right: spherical (3D), planar (quasi-2D) and cylindrical (quasi-1D) distributions of nuclei. The illustrated parameters $R$, $\boldsymbol{n}$, $L$ and $r$ are used in Eqs. (10)-(12). The blue point marks the position of the electron spin.

where the constant $B = |\hbar\mu_0\gamma_e\gamma_n/4\pi| \cdot (4I(I+1)/15)^{1/2}$ is 0.222 MHz $\cdot$ nm$^3$ for protons and $C_{\mathrm{H}}$ [nm$^{-3}$] is the proton number density. We repeated the derivation of $\sigma^2$ for different topologies of proton distributions. For instance, a spin-labelled membrane protein in a protonated membrane in a deuterated solvent can effectively experience a quasi-2D proton distribution. The variance of the hyperfine spectrum is in this case ($R \gg L$)

$$\sigma^2(\infty) - \sigma^2(R) = \frac{5\pi}{8}\left(1 - 3n_z^2 + \frac{27}{8}n_z^4\right)LB^2\frac{C_{\mathrm{H}}}{R^4}. \tag{11}$$

where $n_z$ is the $z$-component of the plane normal vector (Figure 2) and $B$ is defined as in Eq. (10). $C_{\mathrm{H}}$ in the equation above is the proton density in the bi-layer. Repeating the same calculations for quasi-1D-geometry, i.e. a cylinder of radius $r$ and of the axis $\boldsymbol{n} = (n_x, n_y, n_z)^T$, yields

$$\sigma^2(\infty) - \sigma^2(R) = \frac{2\pi}{5}\left(1 - 3n_z^2\right)^2 r^2 B^2\frac{C_{\mathrm{H}}}{R^5}. \tag{12}$$

The one- and two-dimensional geometries are characterized by a steeper dependence on $R$ than the three-dimensional case. Consequently, we expect that the sensitivity range in the ih-RIDME method is closer to the electron spin in lower dimensions as compared to the three-dimensional distribution.

Not all protons around the electron spin contribute to the spectral diffusion processes. The inter-nuclear coupling of close protons is not strong enough to mix the hyperfine levels and induce the flip-flop transition. If in a pair of protons, the difference of the hyperfine couplings is much greater in absolute value than the nuclear-nuclear interaction, we call such protons strongly coupled to the electron. In terms of the nuclear-pair ESEEM model, the ESEEM modulation depth is close to zero.(Jeschke (2023)) In the multi-nuclear spin system, there is no analytical relation between the parameters of the hyperfine- and nuclear Hamiltonians and the probability of nuclear transitions. We suggested using the value of $R$ that fits the experimentally determined values of $\sigma$ as in Eqs. (10)-(12) (depending on the dimension). Such a characteristic value is called a blocking radius, $R_{\mathrm{bl}}$. It represents the region of electron-nuclear distances where protons have the strongest contribution to spectral diffusion. In reality, due to the anisotropy of the hyperfine and nuclear dipolar interactions, the blocking surface may be non-spherical. With the development of methods to analyze the behaviour of multi-nuclear spin systems, it would be possible to do more accurate structure-based predictions of the active zone in ih-RIDME.

The value of the blocking radius depends on the typical distance between the neighbouring protons because this distance affects the nuclear-nuclear coupling constants in $\hat{H}_{\mathrm{dd}}$. The weaker the average nuclear dipolar interaction, the larger the blocking radius. For a fully protonated water-glycerol (1:1 v/v) mixture, we determined experimentally $R_{\mathrm{bl}} = 1.36$ nm.(Kuzin et al.





(2022)) Upon proton dilution, achieved by using deuterated water and glycerol, the radius grows as $R_{\text{bl}} \propto C_{\text{H}}^{-1/3}$. In combination with Eq. (10), this yields a relation

$$\sigma \propto C_{\text{H}}. \tag{13}$$

The proportionality factor $\sigma/C_{\text{H}}$ for the mentioned water-glycerol mixture is $0.0215\,\text{MHz}/(\text{mol}/L)$ if proton concentration is used and $0.0357\,\text{MHz}\cdot\text{nm}^3$ if used as the number density.

To simplify the description of the spin dynamics during the mixing block, as the $\hat{H}_{\text{dd}}$ cannot be omitted now, we use a diffusion-like approximation based on a formalism of magnetization spectrum.(Kuzin et al., 2022) The latter is an ensemble function that specifies the phase evolution of electron spin packets with a certain hyperfine field accounting for their statistical

weights. The magnetization spectrum also depends on the length of the preparation and diffusion blocks ($t$ and $T$, respectively), which is reflected in the notation $\mu_t(\omega_{\text{hf}}, T)$. Given the approximation in Eq. (7), the spin packet with the hyperfine field $\omega_{\text{hf}}$, after the preparation time $t$, starts the mixing block with the phase $\omega_{\text{hf}}t$ which is reflected as

$$\mu_t(\omega_{\text{hf}}, 0) = e^{i\omega_{\text{hf}}t}\rho(\omega_{\text{hf}}). \tag{14}$$

The evolution of this function during the mixing block is approximated by the following differential equation

$$\frac{\partial\mu(\omega_{\text{hf}}, T)}{\partial T} = D\left(\rho(\omega_{\text{hf}})\mu''_{\omega\omega}(\omega_{\text{hf}}, T) - \rho''(\omega_{\text{hf}})\mu(\omega_{\text{hf}}, T)\right) \tag{15}$$

where $D$ [freq$^3$/time] is the spectral diffusion coefficient. After the mixing block, the detection block follows and the ih-RIDME signal is expressed via the magnetization spectrum as

$$R(t; T_{\text{mix}}) = \Re\left[\int_{-\infty}^{+\infty}\mu_t(\omega_{\text{hf}}, T_{\text{mix}})e^{-i\omega_{\text{hf}}t}\mathrm{d}\omega_{\text{hf}}\right]. \tag{16}$$

Here, we use the letter $R$ instead of $V$ to emphasize that a simplified model is used for the transverse evolution.

In the absence of the analytical solution of Eq. (15), we represent $\mu_t(\omega_{\text{hf}})$ in a basis of Hermitian functions

$$\mu_t(\omega_{\text{hf}}, T_{\text{mix}}) = \frac{1}{\sqrt{2\pi}\sigma}\sum_{k=0}^{K}c_t^{(k)}(T_{\text{mix}})He_k\left(\frac{\omega_{\text{hf}}}{\sigma}\right)\exp\left(-\frac{\omega_{\text{hf}}^2}{2\sigma^2}\right) \tag{17}$$

which allows to express $R(t)$ in matrix form

$$R(t; T_{\text{mix}}) = \boldsymbol{c}_0^T\exp(-i\hat{R}\sigma t)\exp\left(\frac{D}{\sigma^3}\hat{\Gamma}T_{\text{mix}}\right)\exp(i\hat{R}\sigma t)\boldsymbol{c}_0 \tag{18}$$

where $\boldsymbol{c}_0$ is a $(K+1)$-element column $(1,0,0,\ldots,0)^T$ and the matrices $\hat{R}$ and $\hat{\Gamma}$ are defined as

$$\left(\hat{R}\right)_{n,k} = \begin{cases} -1, & \text{if } n = k+1 \\ k, & \text{if } n = k-1 \\ 0 & \text{otherwise} \end{cases} \tag{19}$$

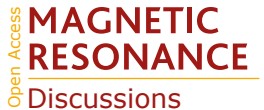

$$\left(\hat{\Gamma}\right)_{n,k} = \begin{cases} 0, \text{ if } n=0 \text{ or } k=0 \text{ or } n+k \text{ is odd} \\ (-1)^k \frac{nk(n+k-3)!!}{n!\sqrt{4\pi}} \left(-\frac{1}{2}\right)^{\frac{n+k}{2}-1} \text{ otherwise.} \end{cases} \tag{20}$$

$R(t;T_{\text{mix}})$ can be also written as a series

$$R(t;T_{\text{mix}}) = \exp\left(-\frac{\sigma^2 t^2}{2}\right) \sum_{k=0}^{K} c_t^{(k)}(T_{\text{mix}})(-i\sigma t)^k \tag{21}$$

where the vector $\boldsymbol{c}_t(T_{\text{mix}}) = \exp\left(\frac{D}{\sigma^3}\hat{\Gamma}T_{\text{mix}}\right)\exp(i\hat{R}\sigma t)\boldsymbol{c}_0$.

We note two key insights from the matrix form in Eq. (18). First, the shapes of the ih-RIDME decays with different $\sigma$ are congruent. There is a family of master decays as functions of the product $\sigma t$ and that is parametrized by mixing time $T_{\text{mix}}$. The decays at a given width $\sigma$ are directly recalculated from these master functions by stretching the argument. Second, the spectral diffusion kinetics is parametrized by $\kappa = D/\sigma^3$ [time$^{-1}$]. So far, there is no *ab initio* derivation of this parameter from the spin Hamiltonian. Nevertheless, it can be interpreted qualitatively. It was found previously (Kuzin et al., 2022) that $\kappa$ is invariant upon isotope dilution of homogeneous solutions. In this case, the density of the nuclear-nuclear coupling network per single nucleus also remains constant. Such density can be interpreted as a measure of the average number of strong homonuclear contacts per nucleus. Should this number be reduced, the kinetics of spectral diffusion slows down and the ratio $D/\sigma^3$ correspondingly decreases. As a model example, a $-(\text{CH}_2)_n-$ fragment in a deuterated solution can be considered. The homonuclear coupling network follows approximately the 1D geometry of the chain and a reduction of the spectral diffusion kinetics is thus expected.

## 2.2 ih-RIDME data model

The ih-RIDME signal in the 5p-RIDME experiment at sufficiently long mixing times has the following product structure

$$V(t;T_{\text{mix}},d_1,d_2) \approx R(t;T_{\text{mix}}) \cdot F(t;d_1,d_2) \tag{22}$$

where $d_1$ and $d_2$ are the static delays of the pulse sequence. The factor $R(t)$ is called a longitudinal factor. It depends only on the position of the mixing block and the mixing time and not on the pulse sequence's static auxiliary delays. This factor was explained in the longitudinal spectral diffusion model described in the previous Section. The second factor $F(t)$ is mixing-time independent and depends on $d_1$ and $d_2$. To give it a quantitative explanation, we need to include the nuclear dipolar interaction to the Hamiltonian during the transverse part of the RIDME sequence

$$\hat{H}^{(\text{prep})} = \hat{H}^{(\text{det})} = \hat{H}_{\text{hf}} + \hat{H}_{\text{dd}} \,. \tag{23}$$

The evolution of the electron spin coherence with this Hamiltonian cannot be solved analytically. By using the perturbation analysis,(Kuzin et al., 2024b) we established some analytical properties of the solution. It was shown that the ih-RIDME factorization as in the Eq. (22) holds if the mixing times exceed the full decay of Hahn echo. If this is not satisfied, the shape of





$F(t)$ becomes mixing-time-dependent. Throughout this work, we refer to $F(t)$ in its limiting form for $T_{\mathrm{mix}} \gg T_m$. The shape
of $F(t)$ has the same congruence property as $R$, i.e. $F$ also depends on the product $\sigma t$.

In heterogeneous systems, every electron spin may be in a different nuclear environment and, hence, have a different hyperfine spectrum. Therefore, the width of the hyperfine spectrum becomes a distributed value described by a distribution function
$p(\sigma)$. The ih-RIDME representation is generalized in this case as

$$V(t; T_{\mathrm{mix}}, d_1, d_2) = \int\limits_{\sigma_{\min}}^{\sigma_{\max}} p(\sigma) V_\sigma(t; T_{\mathrm{mix}}, d_1, d_2) \mathrm{d}\sigma \qquad (24)$$

where $V_\sigma$ is an ih-RIDME trace corresponding to a hyperfine spectrum with the standard deviation of $\sigma$.

### 2.3 Origins of the nuclear heterogeneity

We can indicate several origins of nuclear heterogeneity. The simplest example is fractional heterogeneity, i.e. when a spin probe is distributed between phases or microphases each of which is characterized by a homogeneous proton distribution. This may be a mixture of two solutions with a phase boundary, a biphasic solution of a (bio)molecule prone to condensate
(Emmanouilidis et al., 2021), or paramagnetic sites in a material with different solvent accessibility. In these examples, at each location the proton distribution is homogeneous and the local concentration is thus well-defined but its value is distributed (different at different locations).

A different type of heterogeneity is local (or structural) inhomogeneity. This case can be associated with the conformational flexibility and condensation interaction of spin-labeled macromolecules or a labeled macromolecule in a deuterated
buffer. Upon conformational flexibility, the distribution of the proton environment arises due to the different variants of macromolecule shape, e.g. backbone conformation of a spin-labeled intrinsically disordered protein (IDP). This situation is typical when macromolecules weakly interact forming aggregates or condensates of variable density and homogeneity.(Kuzin et al., 2024a) The proton distribution in such systems can be substantially irregular and a homogeneous proton concentration is thus undefined. Therefore, we can generally use a width of the hyperfine spectrum $\rho(\omega_{\mathrm{hf}})$ to quantify a single proton configuration.
Assuming that the spectral diffusion is ergodic, i.e. that all nuclear spin states are statistically equivalent, the variance of the hyperfine spectrum of a given nuclear coordinate configuration $\mathcal{R} = \{\boldsymbol{r}_j\}_{j=1}^N$ is

$$\sigma^2(R) = \frac{I(I+1)}{3} \sum_{j=1}^{N} A_j^2 \qquad (25)$$

where $I$ is nuclear spin and $A_j$ are hyperfine coupling constants. In the formula above, the quadrupolar interaction for $I > 1/2$ is neglected. The distribution of configurations $\mathcal{R}$, which can manifest in different numbers of nuclei $N$ and their coordinates,
reflects in a distribution of $\sigma$. This principle is illustrated in Figure 3. Since we assume to work in a solid state, the nuclear coordinates do not change. Therefore, the mean value of the hyperfine field for each configuration $\mathcal{R}$ is not observable and we will work with a distribution of standard deviations, $p(\sigma)$.

An anisotropic proton distribution, as it is typical for structural inhomogeneity, obtains a non-zero width in $\sigma$-scale when averaged over all orientations. We can demonstrate this in a numeric experiment. To this end, we generated a spherical proton





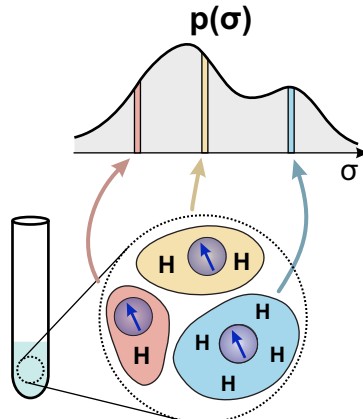

**Figure 3.** Schematic representation of a system with proton heterogeneity and its representation as a distribution of $\sigma$ values related to the local proton densities (see main text for details).

cloud of radius $R$, local concentration $C_{\mathrm{H}}$ and distance from an electron spin $d$ as depicted in Figure 4. Angle $\theta$ parametrizes the position of the cloud in spherical coordinates. Specification of the polar angle $\varphi$ is not required due to the cylindrical symmetry of the problem. This model pictures an anisotropic spatial distribution of nuclei. The distribution of geometric parameters, such as electron-nuclear dipolar angles, is not uniform on a corresponding unit sphere. Consequently, the hyperfine spectrum of the individual configuration is expected to be narrower than that of the purely isotropic one and its shape varies when the configuration is rotated in space. If we expect a Gaussian shape of $\rho(\omega_{\mathrm{hf}})$ in each orientation, then we consider only the variation of its standard deviation.

For the chosen values of $d$ (6, 4, 3 and 2 nm.), we sampled angle $\theta$ over the interval $[0; \pi]$ and computed the standard deviations of the hyperfine spectra with the formula (25). We used in the computation a spherical cloud of $R = 1.5$ nm and $C_{\mathrm{H}} = 20\ M$ which resulted in 170 protons. The hyperfine couplings were calculated using the point-dipole approximation. The resulting distributions $p(\sigma)$ are shown in Figure 4. We can see that for $d = 6$ nm and 4 nm the distributions resemble half of the Pake pattern. This means that the size of the proton cloud is negligible compared to the distance $d$ and the cloud can be approximately represented by a point magnetic moment. Another indication is that the distribution $p(\sigma)$ attains values close to zero. This corresponds to the angles $\theta \approx \Theta_M$ and the scattering of electron-nuclear dipolar angles is again negligible.

The situation changes when the proton cloud is close to the electron and the point approximation for the proton cloud is no longer applicable. This is exemplified in Figure 4 for $d = 3$ nm and $d = 2$ nm where the shape of the distributions deviates from the Pake pattern.

The main conclusion from this computational exercise is that the distributions of proton densities include several sources of broadening that are not possible to *a priori* disentangle. The structural problem in ih-RIDME is more complex than a two-spin interaction in PDS. The PDS signal in the frequency domain is a superposition of Pake patterns with different splitting parameters. Such spectrum, if the underlying distance distribution is sufficiently broad, may look like a smooth feature-less function. Further, a distance between two spins is a clear geometric invariant upon global frame rotation. Consequently, upon





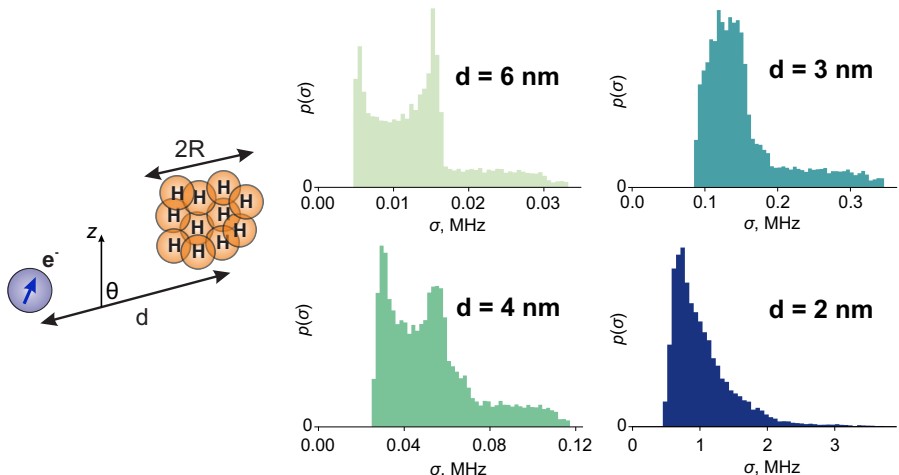

**Figure 4.** A model of proton cloud of radius $R$ with its centre at a distance $d$ from the electron. The histograms are the distributions of the hyperfine spectrum standard deviation ($\sigma$) for different indicated values of $d$. For the simulation, the values $R = 1.5$ nm and $C_{\mathrm{H}} = 20$ $M$ were used.

neglecting the exchange interaction, the dipolar spectrum can be "deconvoluted" from the Pake pattern by various mathematical means (Jeschke et al. (2006); Fábregas Ibáñez et al. (2020); Worswick et al. (2018); Russell et al. (2022)) and the result corresponds to a distance distribution. Since the ih-RIDME method, by design, analyses interaction with many nuclear spins,

the description of a geometric configuration by a single invariant is not feasible. This obstacle affects the use of ih-RIDME data in ensemble-based computation methods, because the intermediate processing result in the form of distribution function $p(\sigma)$ may be less intuitive than the distance distribution in PDS. In the cases where the nuclear distribution around the electron spin is close to isotropic, the value of $\sigma$ can be related to the nuclear density. This is relevant for the first kind of heterogeneity and the function $p(\sigma)$ has the meaning of the distribution of local nuclear density. This meaning is strictly speaking not applicable

to systems with strongly anisotropic nuclear arrangements.

### 2.4   Spectral diffusion beyond protons

The description of spectral diffusion in ih-RIDME driven by the homonuclear coupling applies potentially to any nuclei with non-zero magnetic moment $\mu_{\mathrm{nuc}}$. The relevant candidates are e.g. $^{19}$F ($\mu_{\mathrm{nuc}}(^1\mathrm{H})/\mu_{\mathrm{nuc}}(^{19}\mathrm{F}) = 1.06$), $^{13}$C ($\mu_{\mathrm{nuc}}(^1\mathrm{H})/\mu_{\mathrm{nuc}}(^{13}\mathrm{C}) = 3.98$), $^{31}$P ($\mu_{\mathrm{nuc}}(^1\mathrm{H})/\mu_{\mathrm{nuc}}(^{31}\mathrm{P}) = 2.47$) etc. These are examples of nuclei whose relaxation can be neglected at cryogenic

temperatures on the time scale of a RIDME experiment. The nuclei with a lower gyromagnetic ratio feature slower spectral diffusion kinetics. Indeed, the hyperfine coupling scales as the first power $\mu_{\mathrm{nuc}}$ whereas the homonuclear dipolar coupling scales as a square ($\mu_{\mathrm{nuc}}^2$). Assuming a uniform distribution of nuclei around the electron spin, we obtain the generalization of Eq. (13) (the derivation is presented in Appendix A)

$$\sigma \propto |\mu_{\mathrm{nuc}}|^{11/8} C_n \tag{26}$$





where $C_n$ is the nuclear concentration or density. Since the ih-RIDME kernel depends on the product $\sigma t$, the reduction of $\sigma$ means that longer traces are necessary to achieve the same decay degree. Since, for low-$\gamma$ nuclei, the nuclear dipolar interaction decreases faster than the hyperfine interaction, the diffusion parameter $D/\sigma^3$ is expected to decrease, although the exact dependence is not known. Reduction of $D/\sigma^3$ means that the spectral diffusion effect in ih-RIDME builds up slower with mixing time.

In a mixture of different magnetic nuclei, upon neglecting the heteronuclear dipolar interaction, the ih-RIDME trace is approximated by a product of signals corresponding to each type of magnetic nuclei

$$V(t;T_{\text{mix}}) \approx \prod_j V_j(t;T_{\text{mix}}). \tag{27}$$

Consequently, to better observe the spectral diffusion induced by low-$\gamma$ nuclei, one needs to reduce or eliminate the competitive dynamics from the higher-$\gamma$ nuclei, e.g. by means of isotope replacement or chemical substitution.

## 3 Experimental


The synthesis and characterization of the model compound **1** is presented in the Supporting Information. For our tests, this model compound was dissolved in $H_2O + D_2O + D_8$-glycerol (40% glycerol v/v) to the concentration of 25 $\mu M$ and four samples with bulk solvent proton concentrations of 0 M, 11 M, 22 M and 42 M (Table 1) were prepared. The pulse EPR measurements were performed at Q band ($\nu_{mw} = 34.5$ GHz) at 50 K. The series of RIDME traces were recorded at the

maximum field position with $t_{\pi/2} = 12$ ns, $t_\pi = 24$ ns, with $d_2 = 4.2$ $\mu s$ and for $T_{\text{mix}}$ from 15 $\mu s$ to 480 $\mu s$. For the reference division, traces with $T_{\text{mix}} = 30$ $\mu s$ (Samples **1, 2**) and $T_{\text{mix}} = 15$ $\mu s$ (Samples **3, 4**) were used.

| Symbol | $C_H$(solvent), $M$ | $T_{\text{mix}}^{\text{ref}}$, $\mu s$ |
|--------|---------------------|----------------------------------------|
| **1** | 0 | 30 |
| **2** | 11.4 | 30 |
| **3** | 22.1 | 15 |
| **4** | 41.7 | 15 |

**Table 1.** Composition of solutions of the model compound **1** in water/glycerol (60/40 v/v) to study the solvent contrast.

The pulse EPR measurements in Figure 1c and 11 were done with a solution of TEMPO (2,2,6,6-tetramethylpiperidin-1-yl-oxyl) in a mixture of $H_2O$ and $D_8$-glycerol (60/40 v/v), at Q band at 50 K.





## 4 Discussion

### 4.1 Simplified ih-RIDME kernel

The representation of the ih-RIDME kernel in the exact form is not convenient. We found an approximation for it that allows us to discuss its analytical features. The approximation reads

$$R(t;T_{\text{mix}}) \approx \exp(-\alpha(T_{\text{mix}})\sigma^2 t^2) \tag{28}$$

with

$$\alpha(T_{\text{mix}}) \approx 1 - \exp(-0.245(D/\sigma^3)T_{\text{mix}}). \tag{29}$$

$\alpha(T_{\text{mix}})$ is a growing function from 0 to 1. The quality of this approximation is shown in the SI (Section S2).

In Ref. Kuzin et al. (2022), we showed that the factor $F(t)$ in 5p-RIDME can be approximated as a Gaussian decay

$$F(t) \approx \exp(-\beta\sigma^2 t^2). \tag{30}$$

The shape of $F$ is sensitive to the parameters of the RIDME sequences. In particular, parameter $\beta$ has a weak dependence on delays $d_1$ and $d_2$. So far, the full-Hamiltonian calculation of the transverse factor is not achieved, therefore, we will rely on the model functions approximating it.

By combining Eqs. (28) and (30), we obtain a simplified form for $V_\sigma(t)$

$$V_\sigma(t) \approx \exp(-(\alpha(T_{\text{mix}}) + \beta)\sigma^2 t^2). \tag{31}$$

This approximation implies that the ih-RIDME traces of homogeneous ensembles are close to Gaussian decays. The Gaussian shape is a direct consequence of Gaussian approximation for the hyperfine spectrum (as in Eq. 9). The curvature of the decay $((\alpha(T_{\text{mix}}) + \beta)\sigma^2)$ monotonically increases with mixing time and levels off at high $T_{\text{mix}}$. The rate of the growth is parametrized by $D/\sigma^3$. After the reference division, the parameter $\beta$ is eliminated and the decay curvature is $(\alpha(T_{\text{mix}}) - \alpha(T_{\text{mix}}^{\text{ref}}))\sigma^2$.

In further discussion, we will use a general symbol $K(\sigma t;T_{\text{mix}})$ for the ih-RIDME kernel without specifying whether an approximate or the exact form is taken.

### 4.2 Fitting

The dataset for fitting consists of several ih-RIDME traces measured with different mixing times and a reference trace with the shortest mixing time. The set of mixing times can be chosen as a geometric series with a common ratio of 2, i.e. $T_{\text{ref}}, 2T_{\text{ref}}, 4T_{\text{ref}}, 8T_{\text{ref}}$ etc. The common ratio 2 allows for a simple implementation of a 2D-experiment acquisition using the PulsSPEL scripting language used in commercial Bruker spectrometers. The traces are divided by the reference trace to remove RIDME artefacts that are not covered by the considered model (see SI S4) and fitted by a ratio of computed traces with a test distribution function $p(\sigma)$.



The optimization problem is mathematically formulated as

$$p(\sigma) = \arg\min_{p \geqslant 0} \left\{ \sum_{i=1}^{M-1} w_i \left\| \frac{V(t; T_{\mathrm{mix},i})}{V(t; T_{\mathrm{mix,ref}})} - \frac{\int K(\sigma t; T_{\mathrm{mix},i}) p(\sigma) \mathrm{d}\sigma}{\int K(\sigma t; T_{\mathrm{mix,ref}}) p(\sigma) \mathrm{d}\sigma} \right\| \right\} \tag{32}$$

where $M$ is the number of recorded traces and $w_i \geqslant 0$ are weights. In all further examples, the optimization weights are set to 1. The integration is meant from $\sigma_{\min}$ to $\sigma_{\max}$. Furthermore, the parameters $D/\sigma^3$ and $\beta$ may be additionally optimized. In principle, it is feasible to assume that heterogeneous systems with a broad or multicomponent distribution $p(\sigma)$ may be characterized by a discreet or continuous distribution of these parameters. Here, we do not consider this case and assume single values for $D/\sigma^3$ and $\beta$.

The numeric tests below were performed with a home-written optimization algorithm to facilitate the investigation of the fitting history. The algorithm is written in Python and implements a gradient-descending least-square fitting of a series of ih-RIDME traces.

### 4.2.1 $\delta$-initial condition

We tested the convergence of the fitted distribution in the case of homogeneous systems. Such systems would be described by a single proton concentration, i.e. $\delta$-shape distribution $p(\sigma)$ as in Eq. (33). This test is motivated by noting that, in contrast to the dipolar kernel, the ih-RIDME kernel (see Eq. (31)) is represented by smooth monotonic decays without characteristic minima or maxima. The traces corresponding to close points in $\sigma$-domain are also close in the time domain in the standard function metric. This is a known problem of inverse Laplace transform. However, the ih-RIDME inversion deals with global Laplace inversion of multiple decays, the relative steepness of which is controlled by essentially a single parameter $(D/\sigma^3)$. This condition stabilizes the ill-posed problem. However, as the problem does not become well-posed, an uncertainty of the peak width can be anticipated.

We generated a set of five ih-RIDME traces assuming a certain value of $\sigma = \sigma_0$ and fitted them using a model-free distribution model as in Eq. (24). The time axis of RIDME traces was in 0 to 4 $\mu s$. The mixing times were $T_{\mathrm{mix}} = 60, 120, 240, 480\ \mu s$ and $T_{\mathrm{mix,\,ref}} = 30\ \mu s$. The value of $\sigma_0$ was scanned in a range 0.1 - 1.3 MHz.

$$p(\sigma) = \delta(\sigma - \sigma_0) \tag{33}$$

The resulting normalized fitted distributions are plotted in Figure 5a. We observe that tests with low $\sigma_0$ and high $\sigma_0$ converged to broadened distributions. The mean values are in good agreement with the tested values of $\sigma_0$, as can be seen in the upper plot of Figure 5b. The relative deviation did not exceed 2%. This indicates the high reliability of the distribution's mean values determination. The fitted standard deviation exhibits a minimum at $\sigma_0 \approx 0.64$ MHz and increases at low and high input $\sigma_0$. We can explain this behaviour as follows. The width uncertainty of the low-$\sigma$-region ($\sigma_0 < 0.25$ MHz) is explained by insufficient decay of RIDME traces. In this range of parameters $\sigma$ and $t_{\max}$, the product $\sigma t$ is low and the ih-RIDME decays are effectively quasi-parabolic as $\exp(-x^2) \approx 1 - x^2$ for small $x$. In this case, only the mean value of the distribution $p(\sigma)$ can be precisely obtained but not the full shape. To increase the resolution in this region, one needs to record longer RIDME traces.



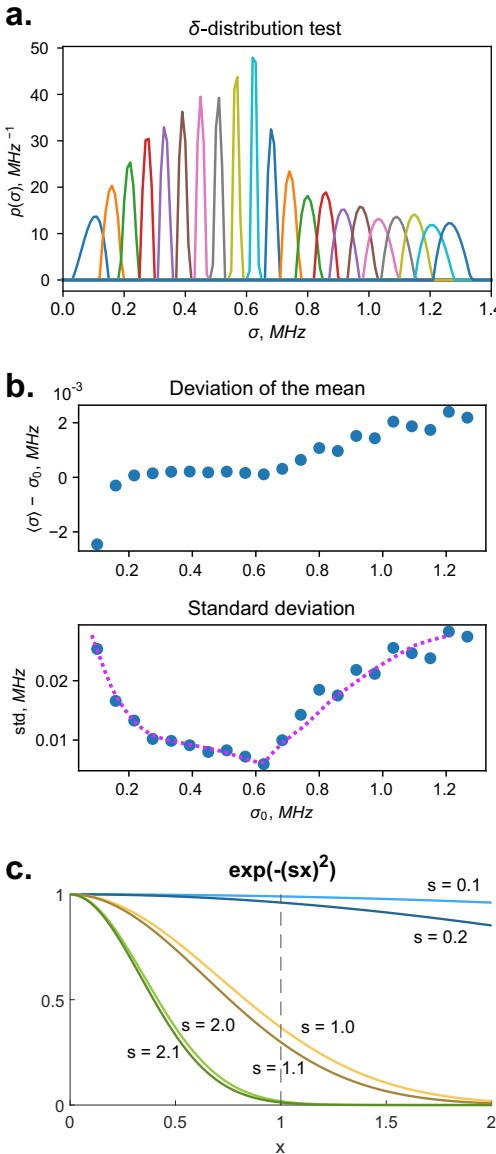

**Figure 5.** (a) Fitted distributions of the $\delta$-condition test. (b) Upper plot: the deviation of the mean value of a fitted distribution from $\sigma_0$ for different tested $\sigma_0$. Lower plot: the standard deviations of the fitted distributions. The dashed line highlights the trend. (c) A plot of three Gaussian decays $\exp(-(s \cdot x)^2)$ of variable $x$ parametrized by $s$. The higher $s$, the lower the shape sensitivity to the shift of $s$. This observation explains the fit broadening of the $\delta$-test at large values of $\sigma_0$ (see text for details).

To illustrate the broadening in the high-$\sigma$-region ($\sigma_0 > 0.8$ MHz), we analyze several Gaussian decays in Figure 5c. There, we plot abstract Gaussian functions $\exp(-(s \cdot x)^2)$ for $s = 0.1$ (light blue), $s = 1.0$ (light yellow) and $s = 2.0$ (light green).
Decays in darker hues correspond to functions where $s$ is incremented by $0.1$. We see that this small increment of $s$ has a weak



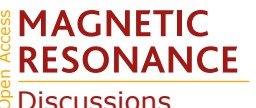

effect at $x < 1$. If the trace length is insufficient (e.g. $x < 1.0$ as we marked by a vertical dashed line to guide the reader's eye) the sensitivity of the trace shape to the change of $s$ is weak. In the range $1 < x < 2$ the trace difference is much larger. This example explains the impact of trace length on the resolution in a low-$\sigma$ domain. The case $s = 1.0$ in Figure 5c shows the highest shape response to the variation of $s$, hence the highest resolution in $s$-domain (analogue of $\sigma$-domain). For $s = 2.0$ and

2.1, the trace difference is again small and this is an intrinsic property since the extension of trace length does not improve the trace distinction due to their decay to zero. Such an observation is reflected in the fit broadening in the high-$\sigma_0$ part.

Overall, we observe a rather good reproducibility of the $\delta$-distributions in the data processing. Global fitting of ih-RIDME data stabilizes the inverse problem. The mean values of fitted distributions can be reliably determined while width uncertainties may remain. In general, given the fixed length of the traces, there is a mid-$\sigma$-region with the highest shape reliability. The size

of this region can be extended into the low-$\sigma$-regions by acquiring longer RIDME traces (i.e. optimizing $t_{\max}$).

### 4.2.2    White Gaussian noise test

We simulated ih-RIDME data assuming a Gaussian distribution $p(\sigma)$ with the centre $\sigma_0 = 0.5$ MHz and standard deviation 0.18 MHz and added Gaussian white noise:

$$V_{\mathrm{sim}}(t; T_{\mathrm{mix}}) = V_{\mathrm{RIDME}}(t; T_{\mathrm{mix}}) + \Delta V(t) \tag{34}$$

where

$$V_{\mathrm{RIDME}}(t; T_{\mathrm{mix}}) = \int\limits_0^1 K(\sigma t; T_{\mathrm{mix}}) p(\sigma) \mathrm{d}\sigma \tag{35}$$

and $\Delta V$ is white noise with a Gaussian distribution of the amplitudes. The noise function was centred ($\overline{\Delta V} = 0$). The noise-to-signal ratio (NSR) of the simulated trace was defined as $\sqrt{\overline{(\Delta V)^2}}/V_{\mathrm{RIDME}}(0; T_{\mathrm{mix}})$. The number of traces and the mixing times were the same as in the $\delta$-distribution test. Figure 6a displays the fitting results with a progressing noise-to-signal ratio.

The original distribution is given as a grey area and lines of different colours correspond to the optimal fits with various noise levels. The noise-free fit almost exactly reproduces the target distribution and the fitting of noised data progressively deviates from it. The ih-RIDME kernel is rather smooth and should work as an effective low-pass filter in the time domain, however, in our tests starting from NSR 0.03 the fitted distributions develop some false shape features.

Due to trace division, the noise level of the reference-divided traces approximately doubles (Figure 6b). Consequently, the

extreme cases of NSR = 0.04 and 0.05 appear to yield a false multicomponent property of the fitted distribution function. This can be suppressed by using derivative-based regularization techniques or others.(Ibáñez and Jeschke (2019)) We applied a penalty function - a norm of the distribution's second derivative - and were able to obtain a fit result close to the original distribution (inset in Figure 6a). For samples with a weak EPR signal, one may opt to record the reference trace with a better quality to reduce noise enhancement upon division or try to fit the ih-RIDME traces without division. The latter option involves

the risk of picking up RIDME artefacts in the fit. However, the influence on $p(\sigma)$ is expected to be within the uncertainty range in the presence of substantial noise. In addition, using model distributions (e.g. mono-Gaussian and multi-Gaussian) is a way





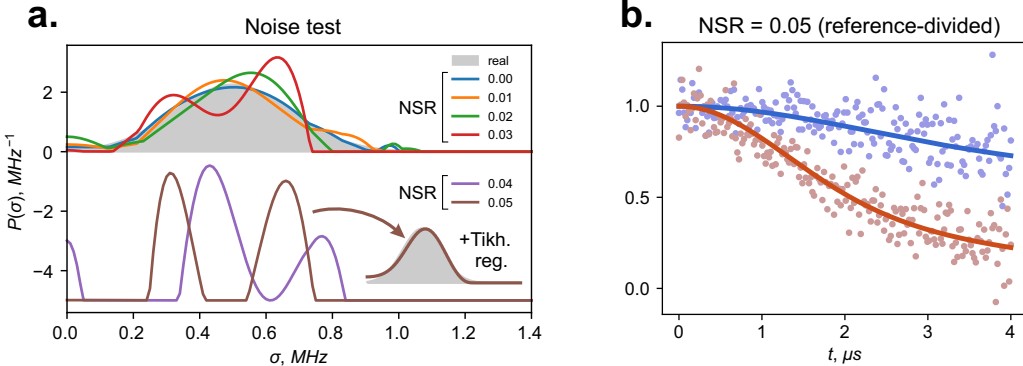

**Figure 6.** Influence of the noise-to-signal level (NSR) in the ih-RIDME traces on the fitting result tested on the generated datasets. a. Deviation of the fit result from the distribution used to generate ih-RIDME traces (grey) at different NSRs. At NSR 0.04 and 0.05 the fitted distribution separates into two components. Tikhonov regularization (adding penalty function of smoothness of the distribution's second derivative) mitigates the splitting. b. Reference-divided traces (dots) and their fits (solid lines) in the case of NSR = 0.05. After the trace division, the effective noise level is notably elevated and the fitted distribution may converge to a distorted shape.

to stabilize the fitting further. As in the case of DEER data analysis (Worswick et al. (2018)) the application of neural networks might also stabilize the solution.

### 4.3 Determination of $D/\sigma^3$ and $\beta$

The optimization problem as in Eq. (32) is parametrized by $D/\sigma^3$ and $\beta$ besides the distribution $p(\sigma)$. We discussed above how these parameters determine the shape of the ih-RIDME kernel $K(\sigma t; T_{\mathrm{mix}})$. $D/\sigma^3$ regulates the dependence of the RIDME decay on mixing time and $\beta$ determines the approximate shape of the $F$-factor. These parameters are sensitive to the properties of an inhomogeneous proton distribution and, therefore, should be optimized for each dataset.

We performed a relaxed scan of $D/\sigma^3$ and $\beta$ of a simulated dataset. For each combination of $D/\sigma^3$ and $\beta$ the pre-generated
dataset was fitted with a model-free distribution with the same number of optimization steps. The analysis of the resulting 2D rmsd-plot (Figure 7(top left and bottom left)) shows that $D/\sigma^3$ can be determined rather certainly, i.e. the 1D-section through the optimal point demonstrates high curvature at the minimum. It turns out that incorrect values of $D/\sigma^3$ have characteristic manifestations in the time-domain fits (Figure 7(top right)). At lower values, the fit outputs an "extended" fork. Oppositely, with the overestimated $D/\sigma^3$, the typical "shrunk" fork is found in the time domain.

In contrast, parameter $\beta$ is characterized by larger uncertainties due to the trace division step. For a homogeneous system, i.e. for $p(\sigma) = \delta(\sigma - \sigma_0)$ for some $\sigma_0$, the $F$-factor cancels exactly making $\beta$ strictly unidentifiable from the reference-divided data. Broad distributions $p(\sigma)$ lead to a weak dependence of the computed reference-divided dataset on $\beta$ as in Figure 7b (green line). For water/glycerol glass in Ref. Kuzin et al. (2022) we found $\beta \approx 0.13$, which can be used as an initial guess in the data analysis.



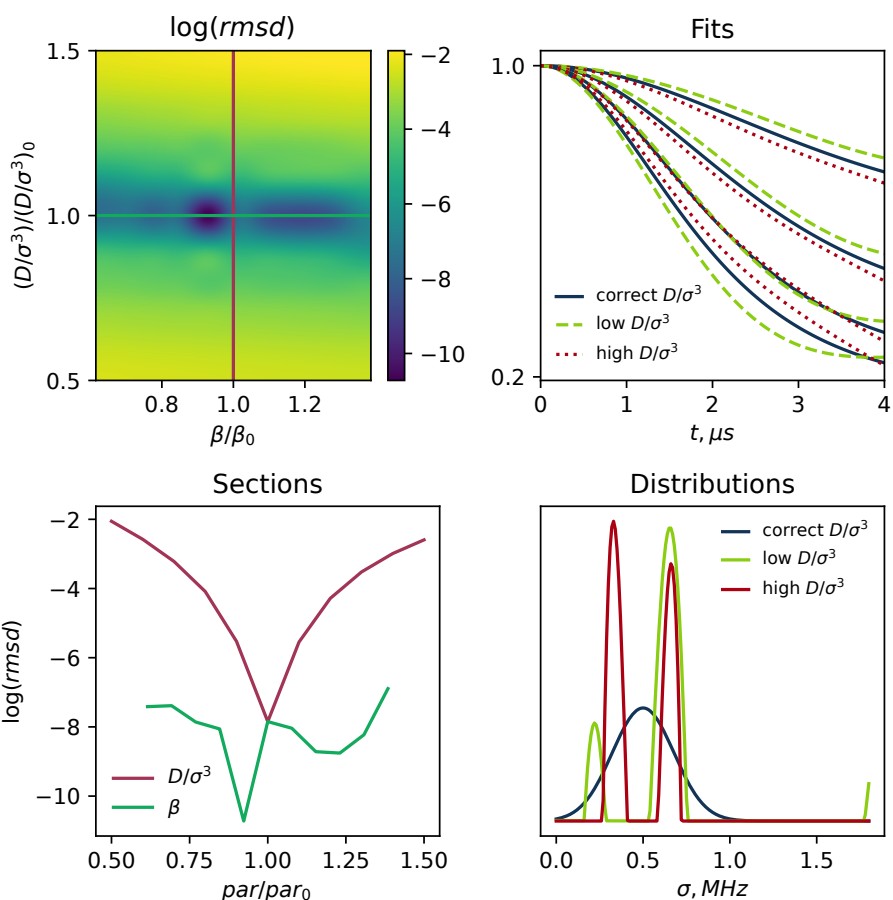

**Figure 7.** Top left: Relaxed scan of $D/\sigma^3$ and $\beta$. The axes correspond to the deviations of the scanned parameters from those used for the dataset generation ($(D/\sigma^3)_0$ and $\beta_0$). Bottom left: Sections of the rmsd plot in the top left panel where one of the parameters is fixed. Top right: Fitting results with a fixed $(D/\sigma^3)$ of low (green dashed line), optimal (navy solid line) and high value (red dotted line) - respectively, $0.5(D/\sigma^3)_0$, $(D/\sigma^3)_0$ and $1.5(D/\sigma^3)_0$. Bottom right: Corresponding fitted local proton concentration distributions.



### 4.4 Test on the model compound 1

#### 4.4.1 ih-RIDME study

We demonstrate the applicability of the above principles on a model compound **1**. It contains the same nitroxide unit as the widely used spin label MTSL. This molecule is also characterized by an anisotropic distribution of protons with respect to the electron spin as will be the case in MTSL-labeled proteins. The chemical structure of the compound **1** is shown in Figure 8a. It contains a shape-persistent linear backbone of conjugated benzene rings and ethynylene units. To this backbone are attached seven polyethylene glycol (PEG) chains. These chains are conformationally highly flexible because of essentially unhindered rotations around C-C and C-O bonds and show no order. The two ethynylene units allow unconstrained dihedral angles between the planes of the benzene rings such that the radical consists of three sections that can freely rotate with respect to each other. Overall, we expect a broad conformational ensemble for this radical.

The fitted model-free distributions are presented in Figure 8b. There, both $C_H$- and $\sigma$-axes are given with a conversion coefficient $0.0215$ MHz/M. The dashed vertical lines mark the proton concentration of the solvents in each sample. We observe that in fully deuterated solvent the distribution of proton densities represents relatively high $C_H$-values (between 10 and 20 M). This reflects that ih-RIDME decay is dominated by the hyperfine interaction with protons of the molecule (PEG sidechains, benzene and triazene rings). The obtained distribution is relatively broad which confirms our expectations from the analysis of the radical geometry.

When the solvent protonation is elevated, we observe two main changes of the $p(\sigma)$, namely, a shift of the mean value and narrowing. The former effect is explained by the increasing number of protons around the electron spin. When the solvent proton density matches the solute proton density, the expectation value of $p(\sigma)$ matches the solvent proton concentration. This is depicted in the correlation diagram in Figure 8d. The deviation of the low-$C_H$ branch from the diagonal due to the protons of the solute is called the solvent contrast effect. The influence of the solvent isotope composition on the distribution width is also a part of this effect. This means that the effective anisotropy of the spatial proton distribution is maximal in the absence of solvent protons.

In practice, full solvent deuteration is not always possible, e.g. if a protonated stock solution is used for a dilution. The solvent protonation degree at which the solvent contrast effect disappears depends on the volume of the studied solute, i.e. on how many solvent molecules are displaced by the (macro)molecule. For a relatively small molecule like **1**, this threshold lies apparently below $C_H = 10$ mol/L. For larger solutes like proteins, protein complexes or biomolecular condensates, characterization of the solute may be possible at higher residual protonation of the solvent.

We also note a solvent effect of a different kind. The optimal $D/\sigma^3$ was found 16 ms in a fully deuterated medium and 24-26 ms in partially protonated samples (Figure 8c). We connect this change with the impact of solvent protons on the connectivity of the proton bath. The main source of protons in compound **1** is the PEG chains. It is plausible to assume that, due to the conformational flexibility and weak interchain interaction, the PEG chains among each other have at most few contacts per chain in an average conformer. This means that all $(-\mathrm{CH_2} - \mathrm{CH_2} - \mathrm{O}-)_n$ chains can be seen approximately isolated, i.e. an average interchain distance is large and the coupling between protons belonging to different chains can be neglected. Therefore,



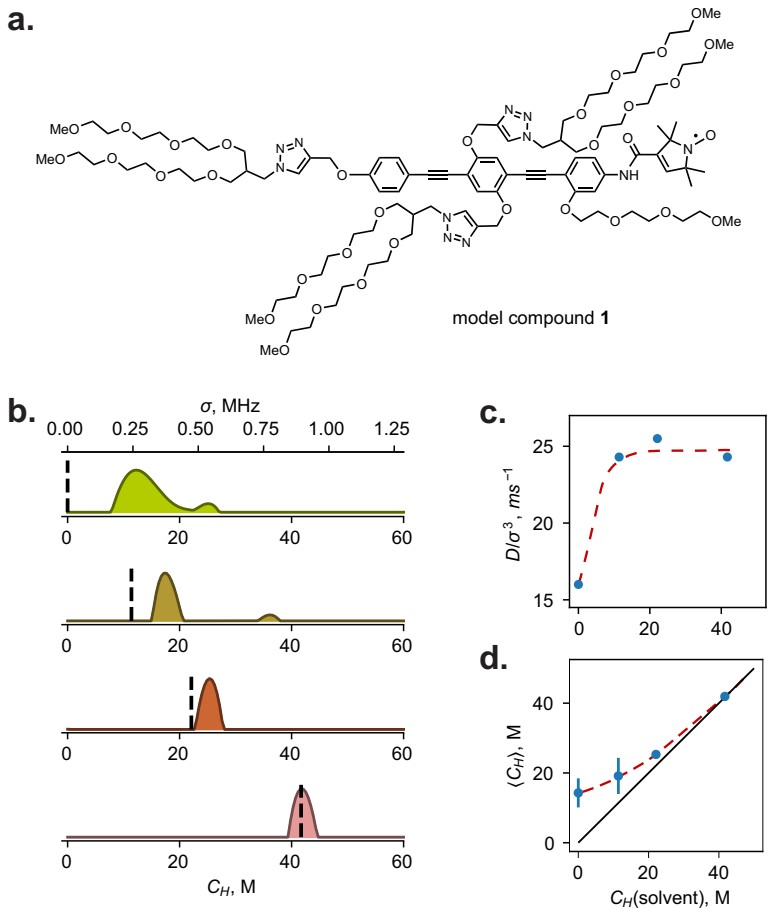

**Figure 8.** (a) Chemical structure of the model compound **1** for testing solvent contrast. (b) Fitted model-free proton density distributions of the compound **1** dissolved in $H_2O$–$D_2O$–$D_8$-glycerol of different protonation degree. Distributions from top to bottom correspond to rows of Table 1. Dashed lines mark the proton concentration of the solvent. (c) The optimal values of $D/\sigma^3$ depending on the solvent proton concentration. (d) Correlation between the solvent proton concentration and the mean proton concentration from ih-RIDME ($\langle C_H \rangle$). The length of vertical lines is equal to the standard deviation of the fitted distributions. Solvent deuteration unveils the internal protons of the solute as the dots deviate from the diagonal line. The dashed lines in panels (c) and (d) show the trend of the parameters change.

the homonuclear flip-flops within the same PEG chain predominate. Solvent protons, being homogeneously distributed, medi-

ate nuclear spin diffusion between the chains and thus enhance spectral diffusion. In particular, the spectral diffusion coefficient $D$ is expected to grow in such a proton configuration compared to the configuration with a quasi-isolated group of protons with a similar value of $\sigma$. Accordingly, the ratio $D/\sigma^3$ is higher. This model agrees with the estimated average inter-proton distance in water/glycerol solvent which is $r_{H,H} = 5$ Å for $C_H = 10$ M. Such distance is smaller than the length of the PEG chain, therefore, the chains can indeed magnetically interact with the solvent protons. Further increase of proton concentration does





not have an effect. Consequently, the value of $D/\sigma^3$ may give information on solvent accessibility or interchain contacts of parts of a molecule, substantially separated from the electron spin.

### 4.4.2  *in silico* analysis

We performed *in silico* analysis of the $p(\sigma)$ distribution. We generated an unrestrained conformational ensemble of the model compound **1** using the Monte-Carlo (MC) approach. For the generation, we used the dihedral angle potentials from Ref. Hoff-
mann et al. (2023). More details on the ensemble generation are given in the SI (Section S5). The ensemble included 1500 conformers.

The generated MC ensemble (Figure 9a) confirmed a significantly anisotropic distribution of the protons. They are concentrated on one side of the nitroxide fragment. The model compound **1** contains 158 chemically unexchangeable protons of which 13 belong to the rigid nitroxide unit. The remaining 145 protons are distributed within a volume of approximately
15 nm$^3$ if the boundaries of the conformational ensemble are approximated as a box. This results in an estimated mean local proton concentration of 16.1 M which is in good agreement with the ih-RIDME data. Nevertheless, the protons are distributed inhomogeneously and form distinct clouds localized around the PEG chains. On the larger scale, the proton ensemble has an ellipsoidal shape with a maximal electron-proton distance of ca. 3.5 nm.

We converted this ensemble into a distribution of local proton densities. For each conformer, we computed the hyperfine
coupling constants with protons in point-dipole approximation. The width of the corresponding hyperfine spectrum $\sigma$ was calculated according to the Eq. (25). The averaging of the molecular orientation was done by sampling the direction of the external magnetic field (thereby, the electron's and nuclear spins quantization axis) over a spherical grid.

To perform the calculation, it is important to determine which protons of the molecule contribute to the spectral diffusion in the ih-RIDME experiment. We performed a simple estimation of the blocking radius by introducing the global cutoff radius
$r_{\mathrm{cut}}$. The protons lying closer to the electron spin than the cutoff are excluded from the nuclear ensemble. We scanned $r_{\mathrm{cut}}$ in the range from 0.5 nm to 2.0 nm and recomputed the distribution of $\sigma$ at each value (see Figure 9b). If close protons are included in the calculation the predicted distribution is shifted to the higher mean values and is broadened. We found that $r_{\mathrm{cut}} = 1.55$ nm leads to the best agreement of the distribution functions based on the rmsd-criterion. At this value, the mean values of the experimental and calculated distributions are also close (see SI S5). The comparison of standard deviation and the skewness
shows a deviation and the reason can be both in the experimental uncertainties and in the ensemble generation. Overall, we conclude that the unrestrained conformational ensemble of compound **1** satisfactorily reproduces the experiment. This could be expected because the model compound **1** does not contain groups with specific interaction, e.g. hydrogen bonding. We tried to improve the agreement by choosing a different force field or by allowing the distribution around the canonical dihedral angles. It was found that these steps had a weak influence on the properties of the conformational ensemble and accordingly
a weak effect on the predicted distribution of local proton density. A further improvement step would be to develop a better physical model for the blocking radius based on the nuclear coordinates. It is feasible to expect that the characteristic size of the blocking radius may be different for the conformers with different spatial properties. Furthermore, in relatively compact structures with high anisotropy, where electron-nuclear vectors are significantly correlated with nuclear-nuclear vectors, the





blocking radius may also depend on the orientation of the molecule in the external magnetic field. Nevertheless, our current

calculation provides an understanding of the sensitivity area of the ih-RIDME method.

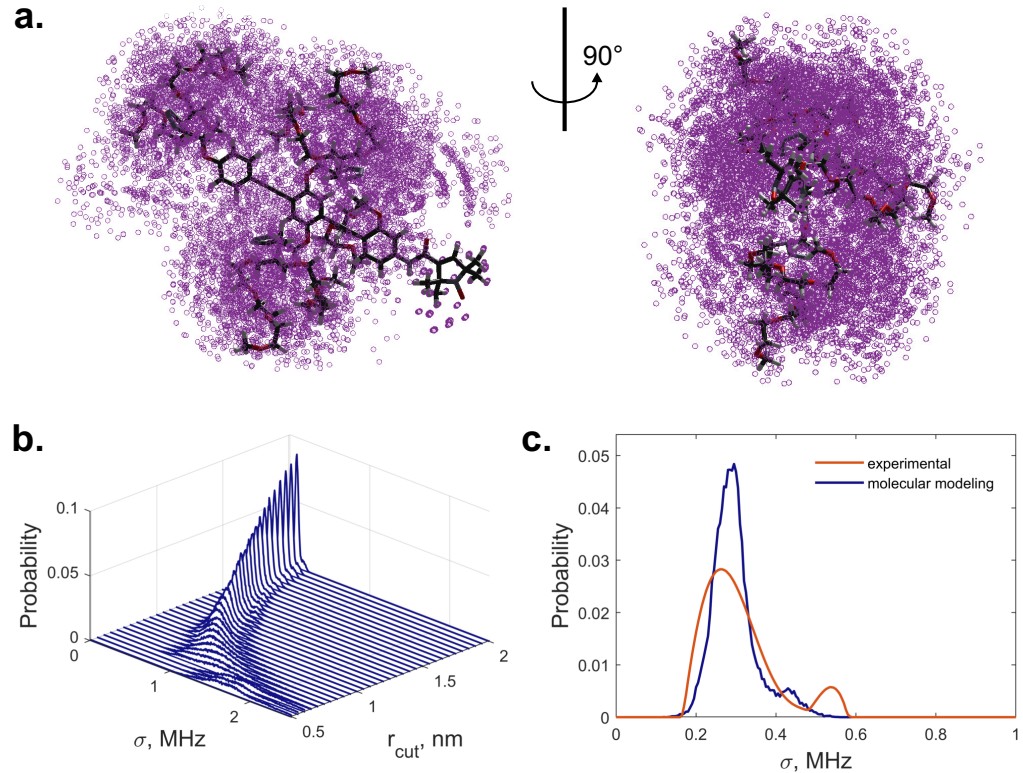

**Figure 9.** *In silico* analysis of the MC-generated conformational ensemble of the compound **1**: (a) a 3D structure and ensemble of 100 randomly selected conformers aligned by the rigid backbone. Magenta circles represent the positions of protons. (b) Calculated distributions $p(\sigma)$ for different cutoffs. (c) The experimental distribution $p(\sigma)$ (orange) and the computed distribution with $r_{\mathrm{cut}} = 1.55$ nm.

### 4.5 Comparison of ih-RIDME sequences

The 5p-RIDME experiment that has been thoroughly investigated so far has some disadvantages and limitations. First of all, the target echo in 5p-RIDME is separated from the beginning of the experiment by a long time, of which $(2d_1 + 2d_2)$ is the transverse evolution that can reach values of 10 $\mu s$ or even more. Due to this, the echo intensity is additionally reduced. Our

numeric tests showed that the noise leads to large uncertainties of the distribution shape. Thus, a long measurement time is needed to accumulate the necessary signal-to-noise ratio (SNR). Besides, if the ih-RIDME method is applied to systems with a substantially broad proton density distribution, the high-concentration fraction may appear attenuated in the experimental data due to $d_2$-filtering. As an extreme, if the $d_2$-delay exceeds the phase memory time of some spin packets by a large factor, these spin packets will not contribute to the ih-RIDME signal and will not be represented in the distribution. On the other hand,

short delays $d_2$ limit the resolution of the low-concentration part as was demonstrated above. This problem is also discussed





for 4p-DEER, which is also a constant-time experiment (Jeschke et al., 2004), and from the application side of RIDME in PDS (Wort et al., 2023). The common principle to reduce or avoid the filtering effects is the use of variable-time experiments.

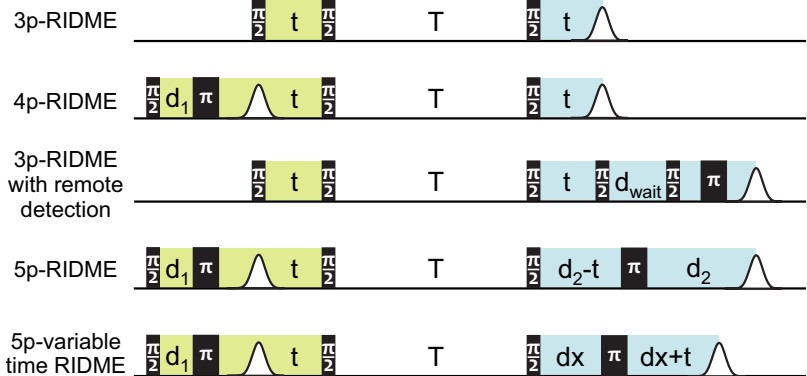

**Figure 10.** Variants of ih-RIDME pulse sequences. In all experiments, the delay $t$ is incremented. The echo in 5p-RIDME is standing and it shifts in all other experiments with the step $2\Delta t$ where $\Delta t$ is the increment step of the delay $t$. The lime and the blue colours highlight the preparation and the detection parts of the sequences (see Figure 1). The phase cycling protocols for the shown sequences are gathered in Appendix B.

We consider four possible analogues presented in Figure 10 and compare them with 5p-RIDME. These sequences have a common mixing block and differ in the preparation and the detection blocks. In 5p-RIDME, the fixed delay $d_2$ limits the available evolution time. In the 3p-RIDME experiment, the traces may be recorded with an arbitrary length, i.e. there is no upper limit on the value of delay $t$ imposed by the pulse sequence. The 3p-RIDME experiment, however, also has disadvantages. First, the three-pulse sequence does not provide an option for $^2$H-ESEEM averaging. Nevertheless, we observed experimentally on various systems that the nuclear modulation cancels out after the division by the reference trace. We suggest using the 4p-RIDME sequence (second sequence in Figure 10), in which the ESEEM averaging is achieved by variation of the first inter-pulse delay, similar to the standard 5p-RIDME.(Keller et al. (2016), see SI S6 for details) Another important issue is the spectrometer dead time that prevents data acquisition at short delays $t$ ($t < 200$ ns at Q-band spectrometers). Due to this, additional uncertainties of the trace normalization can be anticipated. This problem is mitigated by remote detection (3pRD-RIDME, the third sequence in Figure 10). The principle is to apply a mw-pulse with an orthogonal phase at the centre of the echo and thus rotate the magnetization vector to the $z$-axis (i.e. convert coherence to polarization). The length of the magnetization vector does not change and is read out by the last two pulses. The longitudinal waiting time $d_{\text{wait}}$ needs to exceed the spectrometer dead time, but should be short enough so that electron longitudinal relaxation is negligible. The experimental traces of 3pRD-RIDME were measured with $d_{\text{wait}} = 1$ $\mu s$. The delay in the read-out block may be as short as is allowed by the instrumentation. The remaining dead time, in this case, is determined by the properties of the mw-amplifier, namely, the minimal separation time when pulses do not interact with each other. The travelling-wave tube (TWT) amplifier, available to us, allowed for a pulse separation of 32 ns. We also demonstrate experimentally (see SI Section S6) by comparing 3p-RIDME and 3pRD-RIDME that the remote detection block does not distort the ih-RIDME data. Since the trace decay in ih-RIDME is

usually slower than in PDS techniques, the dead-time reconstruction should be more reliable in this case and it can be achieved by extrapolation. We also considered a variable-time RIDME sequence proposed in Ref. Wort et al. (2023) (fifth sequence in Figure 10). The detection principle exploits a single $\pi$-pulse to refocus the virtual echo. The delay $dx$ is fixed and it should be

chosen larger than $d_1$ due to a crossing echo that is not eliminated by a standard 8-step phase cycling protocol. The advantage of this experiment is that it provides both dead-time-free measurement and an option for ESEEM averaging. We refer to this experiment as 5pVT-RIDME.

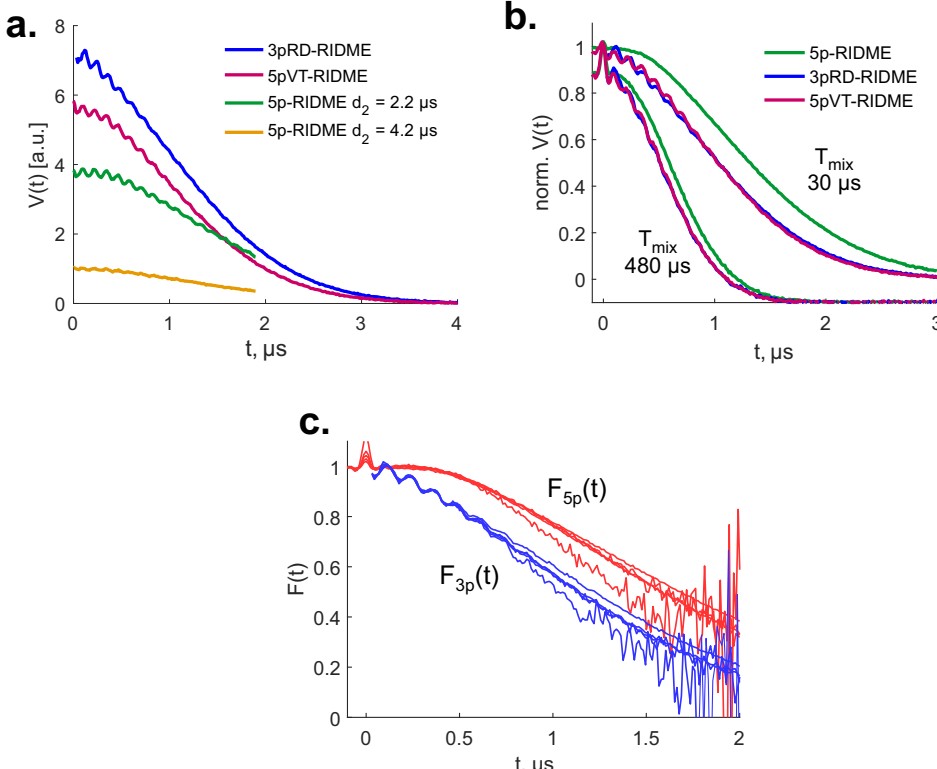

**Figure 11.** (a) Comparison of absolute sensitivity in 3pRD-RIDME (blue), 5pVT-RIDME (red), 5p-RIDME with $d_2 = 2.2\ \mu s$ (green) and $d_2 = 4.2\ \mu s$ (yellow). Mixing time was set to $15\ \mu s$. (b) Comparison of the trace curvature in 5p-RIDME (green), 3pRD-RIDME (blue) and 5pVT-RIDME (red) at $T_{\text{mix}} = 30\ \mu s$ and $T_{\text{mix}} = 480\ \mu s$ (vertically shifted for better visibility). (c) Transverse factors extracted from 3pRD-RIDME (blue) and 5p-RIDME (red).

Next, we compare the sensitivity of 3pRD-RIDME, 5pVT-RIDME and 5p-RIDME. For this, we measured the corresponding traces with the same microwave setup and number of shots per point. Thus, the level of noise is comparable between the

datasets and the comparison of signal intensity represents the comparison of signal-to-noise ratios. For the tests, we measured the 5p-RIDME signal with $d_2$ parameters equal to $2.2\ \mu s$ and $4.2\ \mu s$. The results are presented in Figure 11a. We found that the 3pRD-RIDME signal close to $t = 0$ exceeds both 5p-RIDME signals. The sensitivity enhancement was calculated at $t = 0.08\ \mu s$ and yielded 1.9 for $d_2 = 2.2\ \mu s$ and 7.2 for $d_2 = 4.2\ \mu s$. These values correspond to the reduction of measurement





time by a factor of $\approx 3.5$ and $\approx 52$, correspondingly. The major contribution to such an enhancement stems from the decrease
of the total transverse evolution time. However, less magnetization loss due to fewer pulses in 3p-RIDME also plays a role. It is
discussed theoretically in Ref. Pannier et al. (1999) and shown numerically in Ref. Lovett et al. (2012) that the total excitation
profile of a pulse sequence is narrower with an increased number of mw-pulses. Consequently, fewer spin packets contribute
and the signal intensity is lower. The slight intensity reduction in 5pVT-RIDME compared to 3pRD-RIDME (by factor 1.3
under the conditions of Figure 11a) is most probably explained by the longer total transverse evolution time in 5pVT-RIDME
than in 3pRD-RIDME.

Next, we discuss the data model of the ih-RIDME experiments signal. All sequences in Figure 10 can be split into the
preparation, spectral diffusion and detection parts as it was done for 5p-RIDME. Following the spin dynamics treatment
summarized in the theoretical Section, we know that the ih-RIDME signal is a product of a longitudinal spectral diffusion
factor $R(t; T_{\text{mix}})$, which is independent of the type of the pulse sequence, and the factor $F(t)$ which is determined by the
preparation and detection blocks in a specific pulse sequence:

$$V(t; T_{\text{mix}}, \boldsymbol{d}) \approx R(t; T_{\text{mix}}) \cdot F(t; \boldsymbol{d}) \tag{36}$$

where $\boldsymbol{d}$ denotes an array of all constant auxiliary delays specific to the chosen pulse sequence. For 5p-RIDME $\boldsymbol{d} = [d_1, d_2]$, for
5pVT-RIDME $\boldsymbol{d} = [d_1, dx]$ etc. At the mixing times, which validate the approximation (36), the spin dynamics in preparation
and the detection blocks are uncorrelated and the transverse factor is additionally factorized (Kuzin et al. (2024b))

$$F(t; \boldsymbol{d}) = F_{\text{prep}}(t; \boldsymbol{d}_{\text{prep}}) \cdot F_{\text{det}}(t; \boldsymbol{d}_{\text{det}}). \tag{37}$$

The experimental 3pRD-RIDME and 5pVT-RIDME traces are characterized by a faster decay than 5p-RIDME traces as can
be seen from superimposed normalized data in Figure 11b. Since the longitudinal factor is common in all three experiments,
the transverse factor determines the difference. Consequently, the transverse decay $F_{3p}(t)$ is steeper than $F_{5p}(t)$. In Figure 11c,
we compared them directly by extracting from the experimental traces (see SI S6 for details). Gaussian fit of $F_{3p}(t)$ resulted
in $\beta \approx 0.40$. Slower decay of $F_{5p}(t)$ may be attributed to the role of the last $\pi$-pulse in 5p-RIDME. It partially refocuses the
electron-nuclear interactions and enables a constant transverse evolution time, which balances the decay of $F_{5p}(t)$. The faster
decay of $F_{3p}(t)$ and $F_{5pVT}(t)$, in turn, may cause significantly non-uniform SNR in the reference-divided traces that decreases
fast towards the end of the traces. In principle, new ih-RIDME sequences may be designed by modifying the preparation and
detection parts to optimize the properties of the transverse factor.
Based on the presented comparison, we conclude that 3p-RIDME and 5pVT-RIDME outperform the constant-time 5p-
RIDME and, therefore, can be considered preferable experiments for the ih-RIDME study. In PDS, the steepness of the back-
ground should be minimized to facilitate its separation from the electron-electron dipolar form factor. This is usually achieved
in pulse experiments with a static observer sequence. The variable-time experiments feature additional electron-nuclear decay,
which is why they are less popular in PDS. The signal in ih-RIDME is already determined by the electron-nuclear interactions,
therefore, changing from a constant-time experiment to a variable-time experiment has a weaker effect on the steepness of the
trace decay.





All presented sequences contain a longitudinal block and can be used for the RIDME experiment to detect the electron-electron dipolar interaction. We see a potential for developing and designing the pulse sequences with longitudinal blocks in the hyperfine spectral diffusion phenomenon context. As proposed above, the optimization approach can deal with changing the preparation and detection part. Besides, in principle, the use of multiple mixing blocks can be also investigated. Therefore, we propose HYperfine Spectral Diffusion Echo MOdulatioN (HYSDEMON) as an alternative name for this series of experiments that emphasizes the mechanism dominating the signal generation.

## 5  Conclusions

In the present work, the fundamentals for the analysis of the heterogeneous nuclear ensembles using the ih-RIDME method are considered.

The electron-protons contribution to the electron spin echo decay in the RIDME experiment can be rather well approximated as a result of spectral diffusion of the electron spin within a Gaussian-shaped distribution of the hyperfine field. Accordingly, the width $\sigma$ of such a Gaussian distribution is a measure of the number of protons in the vicinity of the electron spin as well as of the mean distance from the electron spin to the proton cloud. The normalized diffusion coefficient $D/\sigma^3$ in such a description becomes a universal parameter which characterizes the connectivity within the proton cloud and is independent of the position and shape of this cloud. For homogeneous frozen solvent glasses fitting RIDME data can determine differences between characteristic hyperfine spectrum width and characteristic electron spectral diffusion rate between different types of glasses.

RIDME data are recorded as a series of decays corresponding to different mixing times and fitted in a reference-divided form. Such global fitting of multiple traces substantially stabilizes the output of RIDME data analysis. Due to the stability of RIDME data fitting, it is also possible to analyze data for samples with varying densities of protons in the vicinity of paramagnetic centres. The ih-RIDME data is fitted assuming a distribution of $\sigma$ values. Such inhomogeneous proton density distributions correlate with the statistics of interchain contacts of unfolded biopolymers. Thus, the distribution functions can then be converted to structural information.

We investigated and characterized the numeric properties of ih-RIDME data fitting including uncertainties of distribution's mean and standard deviation, robustness to noise, and determination of the parameters of the ih-RIDME kernel. The main results are that the mean value is a reliable output of the fitting routine in a broad range of conditions and determining the distribution's shape features may require optimization of the lengths and the signal-to-noise ratios of the datasets.

These results were further applied to the study of a model compound with a nitroxide, as a source of the electron spin, and with conformational highly flexible proton-rich fragments and substantially anisotropic proton distribution. In a fully deuterated solvent, in the approximation of the rigid cutoff, the blocking radius of 1.55 nm was found. With this radical, we demonstrated two kinds of solvent protonation effects, namely, masking the heterogeneity of the local proton environment and enhancement of the spectral diffusion kinetics mediated by the solvent.





Finally, we considered alternative pulse sequences for the ih-RIDME (HYSDEMON) experiment. While the standard constant-time five-pulse RIDME experiment can introduce additional phase-memory filtering of the proton density distribution function, the three-pulse RIDME and variable-time five-pulse RIDME are free of this effect. In addition, they provide a higher signal-to-noise ratio and thus can be considered as preferential sequences for ih-RIDME (HYSDEMON).

*Data availability.* The experimental data including the EPR and NMR data is available online. DOI: 10.5281/zenodo.14017046.

## Appendix A: Derivation of Eq. (26)

We start from Eq. (10) for a uniform three-dimensional nuclear distribution assuming nuclear density $C_n$ and nuclear magnetic moment $\mu_{\text{nuc}}$

$$\sigma_R = \sqrt{\sigma^2(\infty) - \sigma^2(R)} \propto |\mu_{\text{nuc}}| \sqrt{\frac{C_n}{R^3}} . \tag{A1}$$

The cutoff radius $R_{\text{bl}}$ is approximately determined by the condition that the difference of the hyperfine coupling between the neighbouring nuclei is comparable with their dipolar interaction $|A_j - A_k| \approx \omega_{j,k}$. To obtain qualitative relations, we neglect the angular dependence of both hyperfine and nuclear dipolar interactions. By denoting the minimal distance between the nuclei of the same sort as $d_{\text{min}}$ we obtain

$$\frac{\mu_0}{4\pi\hbar} \frac{3|g_e \mu_B \mu_{\text{nuc}}|}{R_{\text{bl}}^4} d_{\text{min}} \approx \frac{\mu_0}{4\pi\hbar} \frac{\mu_{\text{nuc}}^2}{d_{\text{min}}^3} \tag{A2}$$

where the expression on the left-hand side is the first-order Taylor expansion with respect to the $d_{\text{min}}$. Solving for $R_{\text{bl}}$, we find

$$R_{\text{bl}} \approx \sqrt[4]{3 \left| \frac{g_e \mu_B}{\mu_{\text{nuc}}} \right|} d_{\text{min}} . \tag{A3}$$

In a uniform nuclear bath $d_{\text{min}} \propto C_n^{-1/3}$, consequently

$$R_{\text{bl}} \propto |\mu_{\text{nuc}}|^{-1/4} C_n^{-1/3} . \tag{A4}$$

After substitution of $R_{\text{bl}}$ into Eq. (A1), Eq. (26) is derived.

## Appendix B: Phase cycling protocols for ih-RIDME experiments

In the following Tables, the pulse index (p1, p2 etc.) corresponds to the pulse order in Figure 10. The phases of the not-mentioned pulses are $+x$. The detection (det) means the sign of the echo acquisition.

<table>
<tr><th colspan="4">3p-RIDME</th></tr>
<tr><th>p1</th><th>p2</th><th>p3</th><th>det</th></tr>
<tr><td>+x</td><td>+x</td><td>+x</td><td>+</td></tr>
<tr><td>-x</td><td>+x</td><td>+x</td><td>-</td></tr>
</table>



**3pRD-RIDME**

| p2 | p3 | p4 | det |
|----|----|----|-----|
| +x | +x | +x | + |
| -x | -x | +x | + |
| +y | +y | +x | + |
| -y | -y | +x | + |
| +x | +x | -x | - |
| -x | -x | -x | - |
| +y | +y | -x | - |
| -y | -y | -x | - |

**4p-RIDME**

| p1 | p3 | p4 | det |
|----|----|----|-----|
| +x | +x | +x | + |
| +x | -x | -x | + |
| -x | +x | +x | - |
| -x | -x | -x | - |

**5p-RIDME & 5pVT-RIDME**

| p1 | p3 | p4 | det |
|----|----|----|-----|
| +x | +x | +x | + |
| +x | -x | -x | + |
| +x | +y | +y | + |
| +x | -y | -y | + |
| -x | +x | +x | - |
| -x | -x | -x | - |
| -x | +y | +y | - |
| -x | -y | -y | - |

*Author contributions.* SK and MY conceptualized the project. SK collected and processed the experimental data and performed the numeric tests. SK, GJ and MY discussed the numeric and experimental results. VNS performed the Monte-Carlo simulations and analyzed the conformational ensemble with input from MY. MQ, MF and MH synthesized the model compound **1** under the supervision of AG. The manuscript was written by SK and edited by VNS, AG, GJ, and MY

*Competing interests.* The authors declare no competing interests



*Acknowledgements.* SK acknowledges Dario Stolba for performing preliminary numeric tests of the ih-RIDME fitting. SK, GJ and MY
630   acknowledge funding from the Swiss National Foundation (grant no. 200020_188467).



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
