# Peer review of "ih-RIDME: a pulse EPR experiment to probe the heterogeneous nuclear environment"

_Magnetic Resonance, 2024_

## Author Response (AR1)

**Response to Reviewers**

**Anonymous referee #1**

In this article, the authors analyse the ih-RIDME experiment and the conditions under which it can be utilised to determine proton concentrations or density.

The work introduces the core concepts of ih-RIDME, outlines various scenarios where it is applicable, and evaluates the reliability of the fitting procedure. Additionally, the authors discuss the selection of appropriate pulse sequences.

The study is well-executed and sufficiently novel to merit publication in MR. That said, I do have some recommendations. Firstly, the article is rather lengthy and complex, which makes the key findings difficult to discern and may confuse readers who are not experts in the field.

The derivation of the equations is particularly challenging to follow. Considering the simplicity of the final model after all the simplifications, it might be better to include these derivations as an annex. Furthermore, the multiple geometries explored in the study are ultimately not employed in the analysis and therefore appear somewhat irrelevant to the article's main focus. Simplifying these aspects would enhance readability without undermining the significance of the derivations.

We revised the theoretical background Section excluding the less relevant pieces and moving them to the Appendix or to the Supporting information. In particular, we show the solution to the spectral diffusion equation (Eqs. 17-21 in the original version) in the SI. We shifted the calculations using 2D- and 1D- nuclear distribution geometries (Eqs. 11-12 in the original version) to the Appendix. We also reorganized and revised parts of the manuscript to separate the previous results from the new achievements and to focus on the key topics.

L40-42: do you have a reference?

We based this statement on our previous results and now we include the citation.

L44: "way stronger" could be replaced by significantly stronger?

This paragraph was removed upon the manuscript editing.

The relation between equation 9 and 10 is not obvious, as you only extract a difference in sigma in eq. 10. If the derivation is kept, could you expand and give an example of the derivation of 10?

We included the step-by-step derivation in the Appendix A.

It is also unclear how eq 13 is obtained.

We extended the explanation in the text.

What does the sentence "R instead of V to emphasize that a simplified model" (L 174) brings to the reader? V is nowhere mentioned previously.

We extended this sentence to avoid confusion. We also now explicitly refer in this sentence to the equation where V(t) is first introduced.

Derivation of R and Gamma is nowhere straightforward, which is why I would either expand or keep the derivation in the annex.

The derivation of matrix elements for R and Gamma was published earlier (Kuzin *et al.* PCCP 2022). We added the reference and moved the discussion of R and Gamma to the SI.

Is a "numerical experiment" commonly referred as a simulation? (L239)

Yes, this is a simulation. We updated the wording.

In figure 9(b), the 1.55 nm is difficult to visualise

We edited this Figure by adding a 1-nm scale. We also added a Figure in the SI (Figure S5.7) where all protons within a 1.55-nm sphere in the molecular ensemble are coloured grey.

As a matter of preference, it is worth noting that spin diffusion is not completely blocked (L34); if it were, DNP would not be possible. Several recent DNP studies have demonstrated that spin diffusion remains active, albeit likely slower. For instance, see Pang et al. (10.26434/chemrxiv-2024-zr8zv) or Stern et al. (10.1126/sciadv.abf5735). I would strongly recommend including these findings in your revised manuscript, along with the consideration that spin diffusion may depend on factors such as temperature and electron relaxation times.

We thank the Reviewer for motivating us to deepen the discussion of this topic and suggesting the literature sources. The nuclear spin diffusion is not blocked as evidenced in different DNP/NMR experiments. We note that the conditions, under which this was observed, imply long waiting times (tens of milliseconds to seconds as shown in Stern et al.). The length of the mixing block in ih-RIDME is limited by the electron spin relaxation time (from 500 microseconds to few milliseconds for nitroxide radicals at 50 K). Therefore, on the time scale used in our experiments, the blocked-core approximation is applicable.

The temperature effects are not expected in this work since the spectral diffusion is driven by static, time-independent Hamiltonians rather than by nuclear spin relaxation. The effect of the electron relaxation times is not relevant for spin-diluted systems (e.g. $C_e < 500$ μM).

We included the references and adjusted accordingly the discussion in Section 4.4.2 of the revised manuscript.

**Anonymous referee #2**

This manuscript discusses and evaluates the use of the RIDME pulse EPR method for investigating anisotropic proton distributions around electron spins. ih-RIDME is a relatively new and interesting concept that opens new application fields A comprehensive discussion will clearly help dissemination and uptake by the community. There are a few points that might make the manuscript more accessible and open it to a wider readership.

Major points

1. This work clearly builds on and extends prior work by Kuzin and Yulikov. However, the way the current manuscript is written and the general lack of referencing of prior concepts makes it difficult to follow what exactly is new in this manuscript and what is reproduced from earlier work to provide the relevant context to the reader. It would help if a revised version clearly delineated concepts taken from earlier work and what has been newly derived here. The current manuscript meanders between a tutorial and original research results.

We reorganized parts of the manuscript to separate the overview of the earlier published concepts from the new results. In particular, we shifted the former subsection 2.3 to the Results and Discussion.

2. The purpose of the derivation of eqs 10-12 seems to not be expressed very clearly. The variance of the hyperfine spectrum will depend on the blocking radius more steeply in lower dimensional distributions. However, lower dimensional proton distributions bring fewer protons close to the electron spin effectively reducing the local concentration. How do these tow factors influence the sensitivity of ih-RIDME. Can this sensitivity be defined in a way that is unambiguous with respect to SNR?

Here, we talk about the sensitivity range, i.e. the typical electron spin - nuclear spin distances that still substantially contribute to the variance of the hyperfine spectrum for a certain geometry of the nuclear spins distribution. These equations and their discussion were moved to Appendix A. We added a sentence "*Distant nuclei have smaller contribution to the hyperfine spectrum and variation of their positions has weaker impact on $\sigma$*" to stress the meaning of sensitivity.

3. The theoretical background section does not exhaust the description of all approximations made and all definitions of terms used (e.g., magic angle). The discussion, that looks more like a results and discussion section, starts with an approximation that is not explicitly stated. The accessibility of the manuscript for a wider readership would be helped if all terms are defined clearly, different use of variables clearly stated, and approximations are clearly defined or referenced.

We revised the Theoretical background Section.

4. The ih-RIDME data that is shown in SI2 is concerning. All traces have a sharp modulation feature around zero time that does not eliminate upon reference division. It grows with mixing time and thus appears to be a PDS signal. Indeed, looking at the synthesis of **1** the final step is a Sonogashira-Hagihara reaction. If significant Glaser reaction product from H-EP$_{R4}$-NO is formed, it is not clear how this can be separated or identified. While the collaborative nature of this work is appreciated, it would have been more appropriate to have the synthesis refereed in a journal with some synthetic chemistry scope. Figure S4.3 shows a modulation that would fit the biradical from a homocoupling of H-EP$_{R4}$-NO. There is no proof of purity provided for **1** and it seems no alternative experiment such as an NO-NO DEER has been performed to identify the source of this unexpected signal that does not eliminate with division. How would the proton distribution in this biradical look? Different but anisotropic. In this case, what is the analytical value of the experiments presented here if such a prominent impurity (modulation depth 10-15%) has no effect on the data analysis?

The mentioned sharp feature in the RIDME data is known as a zero-time artefact. It appears due to the echo crossing at $t = 0$, namely, one of the stimulated echoes and a primary echo from last two pulses of the sequence overlap at $t = 0$. This feature was reported earlier (doi: 10.1039/C7CP01524K) and seen in homogeneous solutions of monoradicals (doi: 10.1039/C8CP07815G). This feature does not originate from the electron-electron dipolar interaction. We now discuss of this feature in the main text. We also added the data on the substance purity (see our answer to issue #5) which excludes the biradical admixture.

In regard to the referee's comment that "it would have been more appropriate to have the synthesis refereed in a journal with some synthetic chemistry scope", we would like to add that the synthesis is straightforward, extensively described and all intermediate products are characterized. From our point of view, the requirements for the synthesis reproducibility are satisfied.

5. To reassure readers about point 4, chromatograms from the purification of **1** and mass spectra could have been shown. The Glaser reaction of H-EP$_{R4}$-NO could have been performed as a reference for identifing the unassigned modulation of the ih-RIDME and show the retention of this biradical in the chromatogram of **1**. In Scheme S-1 two different chemical entities are assigned **1**. The first **1** in the last row should be corrected to **14**.

We are aware of Glaser coupling product (oxidative alkyne dimerization product) as a common side product in alkynyl-aryl couplings. This is always a main aspect when deciding on the sequence of adding structural subunits when assembling the overall compound. Accordingly, we combined in the last alkynyl-aryl coupling iodo compound **5**, which brings along three branched PEG side chains, and alkyne **14**, which brings along one linear PEG side chain. The branched PEG side chain influences the retention time on silica gel much more than the linear PEG side chain (see on page 2556 of Qi et al., *J. Org. Chem.* **2016**, 81, 2549-2571). The Glaser coupling product (alkyne dimer) **15** has only two linear PEG side chains, while the targeted coupling product **1** has one linear and three branched PEG-side chains.

We indeed detected the alkyne dimer **15** during preparative HPLC (see Figure S-1 in the revised version). As expected, it was eluted as a well separated fraction at much shorter retention time. Additionally, we checked the purity of the fraction that consisted of model compound **1** with HPLC-MS (see Figure S-2 in the revised version).

We now have added the structural formula of alkyne dimer **15** in Scheme S-1, information of the elution of this compound, the analytical data of this compound, the chromatogram of the preparative chromatography and the result of the purity check of model compound **1** with analytical HPLC. All these data let us exclude that model compound **1** was contaminated with alkyne dimer **15**.

The label in Scheme S-1 was accordingly corrected.

6. For consistency ether SNR or RNS should be used That monomodal distributions are only recovered for SNR 50 and above is a point that must be emphasised when discussing Fig 6. It means that ih-RIDME requires a set of ~5+ traces with an SNR greater 50 making this an experiment that requires substantial instrument time.

We use consistently SNR now. The discussion of Fig 6 is extended.

"*We note that the mean value and the standard deviation of the distribution are stable upon adding noise to the data (see SI S3). Accurate determination of the distribution shape using a model-free approach in this example requires SNR values from 50 or higher.*"

7. The manuscript revolves around anisotropic proton distributions and in the end a distribution is compared to a model. It is unclear how sensitive this is as there are justifiable doubts that the ih-RIDME signal only arises from what is to be believed **1**. Nevertheless, there seems to be no experimental extraction of any anisotropy. Distributions in sigma were extracted earlier and it is unclear what the advance made here really is. This should have been stated explicitly.

Above all, we solved the doubts on the sample purity in responses to comments 4 and 5 of this Referee. Consequently, the ih-RIDME data solely belongs to the model compound **1**, a monoradical of high purity. Next, the anisotropic proton distribution around an electron spin is discussed only as one example of heterogeneous systems. Together with conformational flexibility, anisotropic nuclear distribution is expected to have a large contribution to the experimentally extracted distributions of sigma of spin-labeled macromolecules. We now mention this explicitly in the end of Section 4.1 in the revised version. The anisotropic proton distribution of protons in the model compound **1** means that different molecular orientations with respect to the external magnetic field have different spectroscopic properties and orientation averaging step must be done in the computations of p(σ). We mention this explicitly now.

Regarding the comment "Distributions in sigma were extracted earlier" – we are not aware of prior publications where the model compound **1** was studied by ih-RIDME or equivalent techniques.

We replaced the word "*anisotropic*" by "*heterogeneous*" in the Introduction, updated Figure 8 and made the corresponding additions in the Discussion.

Minor points

L18 "'long-range structure' determination" – EPR has too low resolution to determine structures in the classic NMR, crystallography or microscopy sense.

We use "*long-range structural constraints*" now.

L21 When citing 3 references for 19 F ENDOR I wonder why the seminal paper that invigorated this field has not been included (https://doi.org/10.1002/anie.201908584).

The citation is added.

L28 "exchange interaction" in NMR terminology this would be *J*-coupling, spin-spin coupling or indirect dipole–dipole coupling.

We corrected this.

L33 "Blocked by the gradient of the electron's magnetic field" The respective protons are hyperfine shifted away from the matrix peak. This does not require a gradient.

If two nuclei have the same hyperfine coupling of any magnitude, the homonuclear coupling leads to the efficient mixing of nuclear levels (doi: 10.1016/j.jmro.2023.100094). It is the dependence of hyperfine coupling on distance, i.e., a gradient of the hyperfine field that leads to the shift from the matrix peak and thus to the blocking.

L27-56 review some concepts behind proton spin diffusion/electron spectral diffusion but only cite two references by Kuzin *et al.* There has been significant work around these concepts prior to these references and this should be appropriately referenced.

The literature overview is extended.

When discussing the CP2 sequence (L76) this needs more detail about the sample and solvent protonation as well as pulse sequence timings. As written, this seems to suggest all possible CP2 and RIDME comparisons will look like the curves in Fig. 1c.

We added this information to the experimental section.

When trace division is introduced, there should be appropriate referencing. https://doi.org/10.1021/acs.jpcb.5b02118 has shown this convincingly but earlier work by Astashkin and Savitsky proposed this as well.

The reference is added.

Fig S4.3 sample 2 lower has 6 curves with only 5 legend entries. The change in colour between figures is an unfortunate choice that makes this unnecessarily difficult to follow. Sample 3 seems to have a clipped echo at $t = 0$, this should have been checked.

We fixed the problem in the Figure S4.3 (Figure S5.3 in the revised version) with sample 2. The colours throughout Figures S5.3 and S5.4 are now consistent between different tiles and panels.

The echo clipping was generally checked and prevented in all shown EPR measurements.

There should be one acronym used in the manuscript and if this is to be HYSDEMON the question arises if there is any (resolved) modulation expected?

We would like to keep the name ih-RIDME when it is related to the classical five-pulse version of the experiment. HYSDEMON is a broader concept and it includes ih-RIDME as a special case.

We removed "*HYSDEMON*" from the Conclusions.

We propose the term "modulation" in a general meaning. The approach in (doi: 10.1016/j.jmro.2023.100094) allows to repeat the computation for a RIDME sequence and to obtain an analytical expression with two nuclei, where the echo would be modulated with distinct frequencies. In many-nuclear cases, the theoretical modulations are not resolved due to the destructive interference. This is similar to how the nuclear-pair ESEEM effect features resolved echo modulation for a single pair of nuclei and manifests as a smooth decay in large nuclear spin systems (*same reference*).

---

## Author Response (AR2)

**Comments from the Journal**

1. We noticed that you used scientific abbreviations in the "Short summary" text and kindly ask you to provide at least one written out version. This does not apply to chemical elements.

We rephrased the Short summary without scientific abbreviations

2. Please ensure that the colour schemes used in your maps and charts allow readers with colour vision deficiencies to correctly interpret your findings. Please check your figures using the Coblis – Color Blindness Simulator (https://www.color-blindness.com/coblis-color-blindness-simulator/) and revise the colour schemes accordingly.

The Figures layouts and the colours were revised and checked with Coblis.

**Referee #1**

The article is significantly improved. I do have a few minor comments.

"At short electron-proton distances the proton spin diffusion is blocked by the gradient of the electron's magnetic field."
This is incorrect or should be tamed: this may be correct on the time scale of the ih-RIDME experiment.

We specified the condition for this approximation: "*At short electron-proton distances the proton spin diffusion is* **slowed down** *by the gradient of the electron's magnetic field* **and can be considered inactive on a time scale of a single shot in a pulse EPR experiment**."

"In the formula above, the quadrupolar interaction for is neglected I > ½"
This is not mentioned earlier and as your work is focused on spin ½,. 1H here, you could remove this.

This sentence is removed.

"Not all protons around the electron spin contribute to the spectral diffusion processes"
This has to be justified. You may mean "on the times scale of the RIDME experiment"

"If in a pair of protons, the difference of the hyperfine couplings is much greater in absolute value than the nuclear-nuclear interaction, we call such protons strongly coupled to the electron"
This depends on the T2 ZQ of the nuclei pair and may only be called "blocked" on the time scale of the ih-RIDME.

Since these two remarks are related to the same paragraph, we added a sentence in front of it: "*We restrict further considerations to the cases to the systems and measurement conditions where nuclear longitudinal ($T_{1n}$) and zero-quantum relaxation ($T_{2, ZQ}$) can be neglected on the time scale of the RIDME experiment.*"

"Such a characteristic value is called a blocking radius, Rbl. It represents the region of electron-nuclear distances where protons have the strongest contribution to spectral diffusion"
blocking radius is confusing. I would have called Rsp for spin diff as it is where the spin diffusion is acting, not blocked.

We agree with this suggestion. We changed $R_{bl}$ to $R_{sp}$ and did minute text adjustment for consistency.